# Optimising age coverage of seasonal influenza vaccination in England: A mathematical and health economic evaluation

Edward M. Hill[1]*, Stavros Petrou[2,3], Henry Forster[4], Simon de Lusignan[3,5], Ivelina Yonova[3,5], Matt J. Keeling[1]

**1** The Zeeman Institute for Systems Biology & Infectious Disease Epidemiology Research, School of Life Sciences and Mathematics Institute, University of Warwick, Coventry, CV4 7AL, United Kingdom, **2** Warwick Clinical Trials Unit, Warwick Medical School, University of Warwick, Coventry, CV4 7AL, United Kingdom, **3** Nuffield Department of Primary Care Health Sciences, University of Oxford, Oxford, OX2 6GG, United Kingdom, **4** Government Statistics Service, Department of Health and Social Care, Leeds, LS2 7UE, United Kingdom, **5** Royal College of General Practitioners, London, NW1 2FB, United Kingdom

* Edward.Hill@warwick.ac.uk

**Data Availability Statement:** The GP consultation data and Hospital Episode Statistics (HES) data contain confidential information, with public data deposition non-permissible for socioeconomic

## Abstract

For infectious disease prevention, policy-makers are typically required to base policy decisions in light of operational and monetary restrictions, prohibiting implementation of all candidate interventions. To inform the evidence-base underpinning policy decision making, mathematical and health economic modelling can be a valuable constituent.

Applied to England, this study aims to identify the optimal target age groups when extending a seasonal influenza vaccination programme of at-risk individuals to those individuals at low risk of developing complications following infection. To perform this analysis, we utilise an age- and strain-structured transmission model that includes immunity propagation mechanisms which link prior season epidemiological outcomes to immunity at the beginning of the following season. Making use of surveillance data from the past decade in conjunction with our dynamic model, we simulate transmission dynamics of seasonal influenza in England from 2012 to 2018. We infer that modified susceptibility due to natural infection in the previous influenza season is the only immunity propagation mechanism to deliver a non-negligible impact on the transmission dynamics. Further, we discerned case ascertainment to be higher for young infants compared to adults under 65 years old, and uncovered a decrease in case ascertainment as age increased from 65 to 85 years of age.

Our health economic appraisal sweeps vaccination age space to determine threshold vaccine dose prices achieving cost-effectiveness under differing paired strategies. In particular, we model offering vaccination to all those low-risk individuals younger than a given age (but no younger than two years old) and all low-risk individuals older than a given age, while maintaining vaccination of at-risk individuals of any age. All posited strategies were deemed cost-effective. In general, the addition of low-risk vaccination programmes whose coverage encompassed children and young adults (aged 20 and below) were highly cost-effective.

reasons. The GP consultation data resides with the RCGP Research and Surveillance Centre and are available via the RCGP RSC website (www.rcgp.org.uk/rsc). The HES database resides with NHS Digital and are available via the HES webpage (https://digital.nhs.uk/data-and-information/data-tools-and-services/data-services/hospital-episode-statistics). All other raw data utilised in this study are publicly available; relevant references and data repositories are stated within the main manuscript and Supporting Information.

**Funding:** EMH, SP and MK are supported by the National Institute for Health Research [Policy Research Programme, Infectious Disease Dynamic Modelling in Health Protection, grant number 027/0089]. (https://www.nihr.ac.uk) MK is funded by the Engineering and Physical Sciences Research Council through the MathSys CDT [grant number EP/S022244/1]. (https://epsrc.ukri.org) SP receives financial support as a National Institute for Health Research Senior Investigator [NF-SI-0616-10103]. (https://www.nihr.ac.uk/explore-nihr/academy-programmes/senior-investigators.htm) This report is independent research funded by the National Institute for Health Research (NIHR) [Policy Research Programme, Infectious Disease Dynamic Modelling in Health Protection, 027/0089]. (https://www.nihr.ac.uk) The views expressed are those of the authors and not necessarily those of the NIHR or the Department of Health and Social Care. The funders had no role in study design, data collection and analysis, decision to publish, or preparation of the manuscript.

**Competing interests:** I have read the journal's policy and the authors of this manuscript have the following competing interests: Although not directly related to this study, Simon de Lusignan has received funding through his University for membership of advisory boards for Sanofi and Sequirus, and funding for influenza vaccine studies from GSK and Seqirus. All other authors declare that they have no competing interests.

The inclusion of elder age-groups to the low-risk programme typically lessened the cost-effectiveness. Notably, elderly-centric programmes vaccinating from 65-75 years and above had the least permitted expense per vaccine.

## Author summary

Vaccination is an established method to provide protection against seasonal influenza and its complications. Yet, a need to administer an updated vaccine on an annual basis presents significant operational challenges and sizeable costs. Consequently, policy makers typically have to decide how to deploy a finite amount of resource in a cost-effective manner. A combination of mathematical and health economic modelling can be used to address such a question. Here, we developed an age- and strain-structured mathematical model for seasonal influenza transmission dynamics that incorporates mechanisms for immunity propagation, which we used to reconstruct transmission dynamics of seasonal influenza in England from 2012 to 2018.

We then performed a health economic evaluation assessing the cost-effectiveness of extending a seasonal influenza vaccination programme of at-risk individuals to also include, for targeted age groups, those individuals at low risk of developing complications following infection. The findings suggest the inclusion of low-risk vaccination programmes whose coverage encompassed children and young adults (aged 20 and below) to be highly cost-effective. In contrast, the inclusion of elder age-groups to the low-risk programme typically lessened the cost-effectiveness.

## Introduction

Seasonal influenza inflicts a stark burden on the health system during winter periods in England. Estimates attribute seasonal influenza as the cause of approximately 10% of all respiratory hospital admissions and deaths [1]. The World Health Organization (WHO) has expressed vaccination as the most effective way of preventing seasonal influenza infection [2]. Vaccination can offer some protection against infection for the individual, while also contributing to reduced risk of ongoing transmission via establishment of herd immunity [3, 4].

WHO recommendations for annual vaccination include those individuals at-risk of complications if faced by season influenza infection, children aged between 6 months to 5 years, and elderly individuals (aged above 65 years) [2]. However, policy makers are typically required to base policy decisions in light of operational and monetary restrictions, prohibiting implementation of all recommendations. Consequently, given an objective of allocating limited resources in a cost-effective manner, the use of quality-assured analytical models is advocated [5]. Thus, sourcing evidence via a combination of mathematical and health economic modelling can help underpin vaccination policy.

The appraisal of seasonal influenza vaccination programmes via coupling of transmission model output and health economic analysis has been applied in differing geographical and social contexts. In recent times, there has been a particular focus on the cost-effectiveness of trivalent versus quadrivalent influenza vaccines. Such analyses have been performed within European settings (Finland [6]), the Eastern hemisphere (Australia [7] and Japan [8]) and low-

and-middle income communities in South Africa and Vietnam [7]. Another application of mathematical and health economic models has been to identify optimal target age groups for seasonal influenza vaccination, such as the scale of a mass paediatric influenza vaccination conjoined to an existing influenza vaccination programme [9].

Switching attention to England, there has been a collection of recent work specific to seasonal influenza vaccination policy. Notably, work by Baguelin *et al.* [10] connected virological data from England and Wales to a seasonal influenza transmission model and interfaced with a health economic analysis model [11], whose findings contributed to the recommendation by the Joint Committee on Vaccination and Immunisation (JCVI) to introduce a paediatric seasonal influenza vaccination programme in the United Kingdom [12]. Subsequent studies have analysed the effect of mass paediatric influenza vaccination on existing influenza vaccination programmes in England and Wales [13], evaluated cost-effectiveness of quadrivalent vaccines versus trivalent vaccines [14], and cost-effectiveness of high-dose and adjuvanted vaccine options in the elderly [15].

Nevertheless, seasonal influenza transmission models that have gone on to be interfaced with a health economic model have typically treated each influenza season and each strain circulating within that influenza season independently. Incorporation of a mechanism for the building and propagation of immunity facilitates investigation of the impact of exposure in the previous influenza season, through natural infection or vaccination, on the disease transmission dynamics and overall disease burden in subsequent years. To that end, we enhance our previous non-age structured mechanistic model framework for seasonal influenza transmission, which includes propagation of immunity from one influenza season to the next, through the inclusion of age-structure [16].

In this study, we first present an age- and strain-structured mathematical model for seasonal influenza transmission dynamics that incorporates mechanisms for immunity propagation. The introduced model is used to reconstruct transmission dynamics of seasonal influenza in England from 2012 to 2018. We go on to perform a health economic appraisal of conjectured vaccination programmes in England. In particular, relative to a baseline strategy of vaccination of at-risk individuals alone, we determine threshold vaccine dose prices that achieve cost-effectiveness under several differing paired age band low-risk vaccination schemes. Our findings may be used to convey optimal target age groups for a seasonal influenza vaccination programme in England.

## Methods

### Data overview

Throughout, we define a complete epidemiological influenza season to run from week 36 to week 35 of the subsequent calendar year. We chose week 36 (which typically corresponds to the first full week in September) as the start week to match the start of the epidemic with the reopening of schools.

**GP consultations attributable to influenza.** We derived seasonal rates of primary care consultations per single year age bracket (0yrs, . . ., 99yrs, 100+yrs) per 100,000 population. Consultation rates were attributed to each of the two influenza A subtypes (A(H1N1)pdm09 and A(H3N2)) and the two influenza B lineages (B/Victoria and B/Yamagata).

GP consultation rates were a product of three constituents: (i) Consultations in general practices for influenza-like-illness (ILI); (ii) respiratory virus surveillance; (iii) circulating

strain distribution.

$$\text{GP consultation rate for strain } m \text{ and age } i \text{ in influenza season } y = C_{m,i,y} =$$

$$\text{GP ILI consultation rate for age } i \times \dots$$

$$\text{Proportion of ILI samples influenza positive} \times \dots \tag{1}$$

$$\text{Proportion of influenza viruses in circulation of strain type } m$$

To construct GP consultation rates for ILI in England, we used records provided by the Royal College of General Practitioners (RCGP) Research and Surveillance Centre (RSC). The RCGP RSC network practices monitor activity of acute respiratory infections, with the dataset being nationally representative both demographically and spatially; the demographics of the sample population closely resembling the country as a whole, with the RCGP RSC network of practices spread across England in order to reflect the distribution of the population.

For the influenza seasons 2009/10 until 2017/18 inclusive, RCGP RSC provided weekly, age and risk status stratified records containing the size of the monitored RCGP RSC population, the number of individuals in the monitored population consulting for ILI, and ILI rates per 100,000. For model fitting purposes, these data were aggregated across risk groups to produce weekly ILI rates per 100,000 for each age bracket. We then summed the weekly ILI rates (weeks 36 through to week 35 in the following calendar year) to produce season-by-season ILI rates per 100,000. Expanded information pertaining to the data extraction is provided in Section 1.1 of the S1 Text.

Given not all patients reporting ILI are infected with an influenza virus, virological sample positivity data yield the fraction of ILI reported primary care consultations that were attributable to influenza. We collected relevant data from values within Public Health England (PHE) annual influenza reports (for the 2009/10 to 2017/18 influenza seasons inclusive) [17, 18], and PHE weekly national influenza reports (for the 2013/14 influenza season onwards) [19]. For full details, see Section 1.2 of the S1 Text.

To inform the circulating strain distribution in each influenza season we used publicly available data for the United Kingdom from FluNet [20], a global web-based tool for influenza virological surveillance (additional details are provided in Section 1.3 of the S1 Text). For each age bracket, we applied the circulating strain composition quantities to the overall estimate of ILI GP consultations attributable to influenza, disaggregating the estimate for all influenza strains of interest into strain-specific quantities. In the absence of data breaking down how the strain composition was split between differing ages, we made a simple assumption of the strain composition being applied homogeneously across all ages. We postulate that incorporation of relevant age-stratified data would permit refinements to the parameter inference scheme to capture any heteorgeneities across ages.

These computations resulted in seasonal rates of GP consultations attributable to the two influenza A subtypes and two influenza B lineages per year of age (per 100,000 population). Due to all samples identified as influenza B within the 2009/10 and 2011/12 influenza seasons being untyped, though model runs were initiated from the beginning of the 2009 influenza season, we restricted fitting comparisons to the time span 2012/13 to 2017/18 inclusive (Fig 1).

**Vaccine uptake and efficacy.**   For the 2009/10 influenza season, we acquired vaccine uptake profiles from survey results on H1N1 vaccine uptake amongst patient groups in primary care [21]. For the 2010/11 influenza season onward we collated vaccine uptake information from PHE official statistics [18, 19].

We took central estimates of age adjusted vaccine efficacy for each historical influenza season (2009/10–2017/18 inclusive) from publications detailing end-of-season seasonal influenza vaccine effectiveness in the United Kingdom [22–29]. In influenza seasons where equivalent

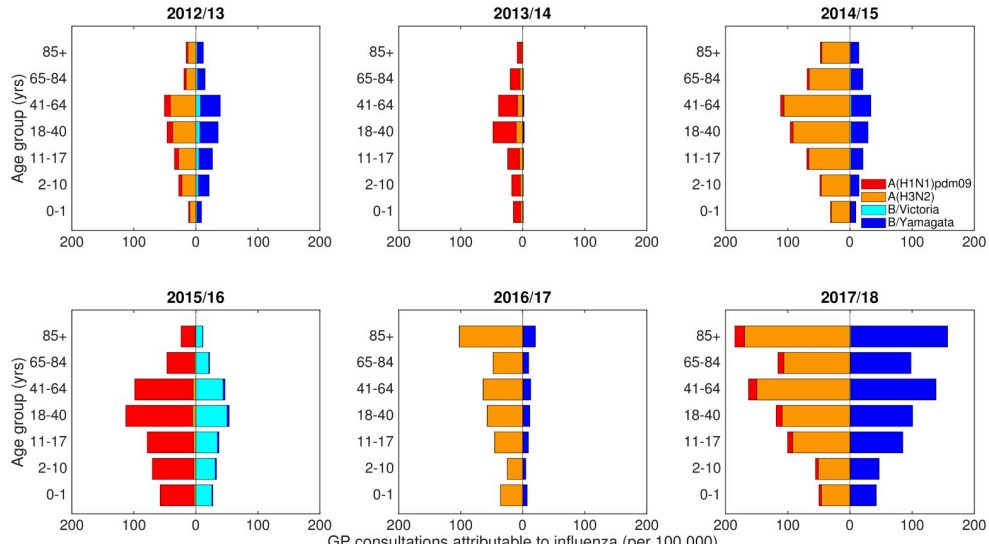

**Fig 1. Empirical, age- and strain-stratified data for ILI GP consultations attributable to influenza per 100,000 population.** The panels cover influenza seasons 2012/13 to 2017/18 (inclusive), the time span for which we performed parameter inference, with panel titles stating the respective influenza season displayed. In each panel, stacked horizontal bars in the left-half depict the cumulative total of ILI GP consultations attributable to type A influenza per 100,000 population per age group (red shading denoting the A(H1N1)pdm09 subtype, orange shading the A(H3N2) subtype). In the right-half of each panel stacked horizontal bars present similar data for type B influenza (cyan shading denoting the B/Victoria lineage, dark blue shading the B/Yamagata lineage). Note that rates were computed for each single year of age, with displayed bars per age group computed by averaging the individual rates over the specified age ranges.

publications were not available, we used mid-season or provisional end-of-season age adjusted vaccine efficacy estimates from PHE reports [30, 31]. As a sole exception, for the 2009/10 influenza season we used the pandemic vaccine (rather than the seasonal influenza vaccine efficacy) to inform efficacy against A(H1N1)pdm09, with the effectiveness against all other strain types set to zero [22].

The process for acquiring vaccine uptake profiles and the stated strain-specific, vaccine efficacy point estimates used within our study (see Table 1) is described in Sections 1.4-1.5 of the S1 Text.

We recognise the use of a point estimate for a quantity with inherent uncertainty surrounding its value is a simplification. Were vaccine efficacy values to be drawn from a distribution, we would expect additional variability in model projections. However, as the point estimates reflect what was deemed as the likely vaccine efficacy given the available data, we take its usage as a reasonable assumption and its application also aided parameterisation of other facets of complexity contained in the model.

## Dynamic transmission model for seasonal influenza

We built upon a previously described dynamic transmission model for seasonal influenza, which incorporates multiple mechanisms linking prior exposure to influenza virus (in the previous influenza season) to immunity in subsequent influenza seasons [16].

At its core, the previously outlined model takes the form of a deterministic continuous-time set of ordinary differential equations (ODEs), which determines the within-season epidemiological dynamics, and a discrete-time map, informing the propagation of immunity from one season to the next. The main distinction between the pre-existing model and the model

**Table 1. Age adjusted, influenza vaccine efficacy point estimates (by influenza season, age and strain type).** All estimates are presented as percentages. The empirical adjusted influenza vaccine efficacy estimates by influenza season and strain type (presented in the S1 Text, Table A) did not provide individual vaccine efficacy estimates for each influenza A subtype and influenza B lineage. We therefore implemented a series of assumptions to produce the strain-specific, vaccine efficacy point estimates used within our study (described in Section 1.5 of the S1 Text).

| Influenza season | Ages | Vaccine efficacy % (95% CI) | | | | Source |
|---|---|---|---|---|---|---|
| | | A(H1N1)pdm09 | A(H3N2) | B/Victoria | B/Yamagata | |
| 2009/10 | All | 72.0 | 0.0 | 0.0 | 0.0 | [22]* |
| 2010/11 | 0–4 | 87.0 | 87.0 | 47.0 | 0.0 | [23] |
| | 5–14 | 84.0 | 84.0 | 75.0 | 0.0 | |
| | 15–44 | 46.0 | 46.0 | 61.0 | 0.0 | |
| | 45–64 | 52.0 | 52.0 | 45.0 | 0.0 | |
| | 65+ | 74.0 | 74.0 | 67.0 | 0.0 | |
| 2011/12 | 0–4 | 52.0 | 52.0 | 92.0 | 92.0 | [24] |
| | 5–14 | 69.0 | 69.0 | 92.0 | 92.0 | |
| | 15–44 | 7.0 | 7.0 | 92.0 | 92.0 | |
| | 45–64 | 11.0 | 11.0 | 92.0 | 92.0 | |
| | 65+ | 48.0 | 48.0 | 92.0 | 92.0 | |
| 2012/13 | 0–4 | 73.0 | 26.0 | 51.0 | 51.0 | [25] |
| | 5–14 | 73.0 | 26.0 | 74.0 | 74.0 | |
| | 15–44 | 83.0 | 40.0 | 68.0 | 68.0 | |
| | 45–64 | 90.0 | 32.0 | 34.0 | 34.0 | |
| | 65+ | 0.0 | 0.0 | 65.0 | 65.0 | |
| 2013/14 | All | 61.0 | 61.0 | 61.0 | 61.0 | [30]† |
| 2014/15 | 0–1 | 34.5 | 30.3 | 40.4 | 40.4 | [26] |
| | 2–17 | 30.4 | 29.4 | 59.4 | 59.4 | |
| | 18–44 | 34.5 | 30.3 | 40.4 | 40.4 | |
| | 45–64 | 32.4 | 31.1 | 49.2 | 49.2 | |
| | 65+ | 30.2 | 32.6 | 0.0 | 0.0 | |
| 2015/16 | 0–1 | 59.8 | 59.8 | 45.9 | 45.9 | [27] |
| | 2–17 | 48.5 | 48.5 | 76.5 | 76.5 | |
| | 18–44 | 59.8 | 59.8 | 45.9 | 45.9 | |
| | 45–64 | 58.6 | 58.6 | 65.0 | 65.0 | |
| | 65+ | 56.1 | 56.1 | 0.0 | 0.0 | |
| 2016/17 | 0–1 | 38.5 | 36.6 | 52.1 | 52.1 | [28] |
| | 2–17 | 63.3 | 57.0 | 78.6 | 78.6 | |
| | 18–64 | 38.5 | 36.6 | 52.1 | 52.1 | |
| | 65+ | 0.0 | 0.0 | 17.2 | 17.2 | |
| 2017/18 | 0–1 | 69.1 | 0.0 | 18.2 | 18.2 | [29]‡ |
| | 2–17 | 90.3 | 0.0 | 60.8 | 60.8 | |
| | 18–64 | 69.1 | 0.0 | 18.2 | 18.2 | |
| | 65+ | 66.3 | 16.8 | 13.2 | 13.2 | |

Data not specified denoted by —.

For influenza seasons where TIV were in use (rather than QIV), N/A in an influenza B field corresponds to the lineage absent from the TIV composition and no efficacy estimate was given (we assume to be 0%).

*: The adjusted seasonal influenza vaccine efficacy in the 2009/10 influenza season was -30% (-89%, 11%) [22]. We therefore used the pandemic vaccine to inform efficacy against A(H1N1)pdm09, with the effectiveness against all other types set to zero.

†: Mid-season estimate of seasonal influenza vaccine effectiveness from [30]. Low incidence throughout the 2013/14 influenza season meant reliable end-of-season estimates for the vaccine efficacy could not be attained (Personal communication, Public Health England).

‡: Provisional end-of-season influenza vaccine effectiveness results.

described here was the inclusion of age-structure. Our age stratification comprised of single year age brackets. We gathered age-partitioned population distribution estimates for England, for the calendar years 2010–2018 inclusive, from Office of National Statistics (ONS) mid-year population estimates [32, 33].

Broadly speaking, the model conjoined four components: a vaccination model, an immunity propagation model, an epidemiological model and an observation model (for a visual schematic of the interactions within and between each model component, see Fig. C in S1 Text). The model generates a sample of epidemiological scenarios for each of the four strains targeted by the quadrivalent seasonal influenza vaccine, namely, two influenza A subtypes, A (H1N1)pdm09 and A(H3N2), and two influenza B lineages, B/Victoria and B/Yamagata. The multi-strain model construction permits exploration of cross-reactive immunity mechanisms and their impact on the disease dynamics.

We retained an assumption of multiple infections per influenza season and/or co-infection events not being permitted. In other words, it was presumed that individuals may only be infected by one strain of influenza virus per season, analogous to natural infection eliciting short-term cross immunity to all other strain types [34].

Next, we detail the differences introduced through inclusion of age-structure versus the earlier described transmission model setup, with additional model details provided in Section 2 of the S1 Text. As we summarise each aspect of the framework, though the model incorporates numerous complexities we also make simplifying assumptions in order to be able to parameterise the model. We highlight these assumptions as appropriate, with potential influences on model outcomes outlined within the discussion.

**Vaccination model.**   Epidemiological states were compartmentalised based on present season vaccine status (indexed by $N$ for non-vaccinated or $V$ for vaccinated). The two facets (within the vaccination model) that play a role in shaping the overall transmission dynamics were uptake and efficacy. Both of these elements were age-dependent.

To inform vaccine uptake, we obtained weekly, time-varying vaccine uptake rates by age $i$, $v_i$. We assumed the rate of vaccination $v_i$ to be constant over each weekly period.

Vaccine efficacy depended upon the extent to which the strains in the vaccine matched the circulating strain in that influenza season (Table 1). We maintained a 'leaky' vaccine, offering partial protection to every vaccinated individual, thus acting to reduce the overall susceptibility of the given group receiving vaccination [35]. Additionally, we assumed those administered vaccines had unmodified transmission (i.e. for those infected, those receiving the 'leaky' vaccine were equally as infectious as those who did not).

**Immunity propagation model.**   Models including explicit immunity propagation mechanisms grant potential for exposure to influenza virus in the previous influenza season, through natural infection or vaccination, to modulate current influenza season susceptibility. We carry over the immunity propagation model framework from the non-age structured variant of the model and apply it independently within each single year age bracket.

In brief, three susceptibility modifying factors of interest were: (i) modified susceptibility to strain $m$ given infection by a strain $m$ type virus the previous season, denoted $a$; (ii) carry over cross-reactivity protection between influenza B lineages, denoted $b$ (to account for infection with one influenza B virus lineage being potentially beneficial in protecting against subsequent infection with either influenza B virus lineage [36]); (iii) residual strain-specific protection carried over from the prior season influenza vaccine, denoted $c_m$. We assumed $0 \leq a, b, c_m \leq 1$, with 0 corresponding to complete protection and 1 corresponding to no reduction in susceptibility.

Furthermore, we let the immunity propagation due to vaccination in the previous season be linearly tied to the age- and strain-specific vaccine efficacy in the previous season

$(\alpha_i^m(y-1))$. In particular, we introduce a linear scaling factor $\xi \in (0, 1)$, reducing the level of vaccine derived immunity between seasons: $c_m(y) = 1 - \xi\alpha_i^m(y-1)$. If $\xi = 0$, there would be no carry over of prior influenza season vaccine efficacy and, therefore, no reduction in susceptibility (with $c_m(y) = 1$). On the other hand, if $\xi = 1$, the full efficacy of the previous influenza vaccine toward strain $m$ was retained. Whilst defining a relationship between two factors when sufficient data are not available to ground it on empirical observation necessitates making assumptions, we recognise a linear dependency is a strong generalisation. Thus, if there is sufficiently detailed data to underpin the models, we encourage consideration of alternative non-linear formulations for the vaccine-derived immunity propagation process. Such formulations may also permit instances of negative interference, where exposure in the prior influenza season (be that naturally or through vaccination [37–39]) increases susceptibility to influenza infection during the current influenza season above the baseline value of 1.

We let $f(h, m)$ denote, for those in exposure history group $h$, the susceptibility to strain $m$. The result was a collection of ten exposure history groupings and associated strain-specific susceptibilities, which we consolidated into a single susceptibility array (see Section 2.2 of the S1 Text). All three propagation parameters $(a, b, \xi)$ were inferred from epidemiological data.

**Epidemiological model.**   At the outset of each influenza season, we initialised the epidemiological model using outputs from the vaccination model (uptake and efficacy) and immunity propagation model. Additionally, we fed in virus transmission rates and the initial proportion of the population infectious per strain.

The within-season transmission dynamics were then performed by a deterministic ODE, age-dependent (single year age brackets), multi-strain structured compartmental based model capturing influenza infection status (with susceptible-latent-infected-recovered, SEIR, dynamics) and vaccine uptake. Latent and infectious periods were age independent, with estimates gleamed from the literature [40, 41] (see Table 2). The ODE equations of the SEIR epidemiological model are given in Section 2.3 of the S1 Text.

Iterating upon the earlier described, non-age structured model variant [16], the inclusion of age meant we could categorically express within the model an age-dependent contact structure and an age-dependent susceptibility profile.

**Table 2. Overview of model parameters.**

| Description | Notation | Value | Sources |
|---|---|---|---|
| **Fixed parameters** | | | |
| Rate of latency loss, influenza A subtypes (day$^{-1}$) | $\gamma_{1,A}$ | $\frac{1}{1.4}$ | [41] |
| Rate of latency loss, influenza B lineages (day$^{-1}$) | $\gamma_{1,B}$ | $\frac{1}{0.6}$ | [41] |
| Recovery rate (day$^{-1}$) | $\gamma_2$ | $\frac{1}{3.8}$ | [40] |
| **Time-varying parameters** | | | |
| Vaccination rate (for age $i$) | $v_i(t)$ | — | [18, 19, 21] |
| Vaccine efficacy (by strain $m$, age $i$, season $y$) | $\alpha_{m,i}^y$ | — | [22–31] |
| **Inferred parameter description** | **Notation** | **Permitted range** | |
| Basic reproduction number, influenza virus strain $m$ | $R_{0_m}$ | $\mathcal{U}(1, 3)$ | |
| Modified susceptibility given natural infection in prior season | $a$ | $\mathcal{U}(0, 1)$ | |
| Modified susceptibility due to type B influenza cross-reactivity | $b$ | $\mathcal{U}(0, 1)$ | |
| Proportion of prior influenza season vaccine efficacy carried over | $\xi$ | $\mathcal{U}(0, 1)$ | |
| Susceptibility by age group $i$ | $\sigma_i$ | $\mathcal{U}(0, 1)$ | |
| Reference ascertainment probability (ages 100+) in season $y$ | $\epsilon_y$ | $\mathcal{U}(0, 1)$ | |
| Relative ascertainment factors (by age $i$) | $\tilde{\epsilon}_i$ | $\mathcal{U}(0, 1)$ | |

We incorporated data on social contacts using estimates for the United Kingdom generated by Fumanelli *et al.* [42], who simulated a population of synthetic individuals in order to derive frequencies of total contacts by age. The transformation of these data into an average number of contacts a given person of age $i$ had with people of aged $j$ are described in Section 2.3.1 of the S1 Text.

With regards to capturing heterogeneity in susceptibility, we considered a temporally independent, age-dependent susceptibility profile. For purposes of model parsimony, the profile took the form of a step function, where we denote $\sigma_j$ for the average susceptibility of age group $j$. As susceptibility had to be inferred, we chose to limit the model to four age bands to avoid overfitting. We considered an average susceptibility for children (0–17 yrs old, $\sigma_{0-17}$), younger adults (18–64 yrs old, $\sigma_{18-64}$), elder adults (65–84 yrs old, $\sigma_{65-84}$), and the eldest adults (85+ yrs old, $\sigma_{85+}$). Further, we imposed a constraint that susceptibility of the younger adults category (18–64 yrs old) could not exceed the susceptibility of any other age band. We acknowledge, at this juncture, that we made a trade-off in the pursuit of parsimony by letting the younger adults category have a notable breadth in age range, which may lead to averaging over heterogeneities.

We assumed that at the outset of each simulation a small fraction of individuals in each age class were infectious. The remainder of the population began susceptible, with no portion of the population initialised with residual immunity. For the 2009/10 influenza season, only the A(H1N1)pdm09 strain was present in our simulation (in accordance with the strain composition data, where proportions of samples for the remaining three strains were low, Fig. B in S1 Text). Thus, we initialised the ODEs such that the initial proportion of the population infectious was set to $1 \times 10^{-5}$ (one per 100,000), partitioned across age groups according to the age distribution. In all subsequent influenza seasons, with all four influenza virus strains being present, the initial proportion of the population infected by each strain was $2.5 \times 10^{-6}$ (with initial cases again partitioned across the age groups according to the age distribution). The initialisation of age-group specific residual immunity, $f(h, m)$, in influenza seasons from 2010/11 onward was governed by a between-season immunity mapping (a mathematical description is provided in Section 2.3.2 of the S1 Text).

Whilst we did not include explicit within-season demographic processes, we did need to account for the witnessed minor changes to the age distribution from one influenza season to the next [32]. We therefore invoked alterations to the population structure (as a result of demographic processes) using a discrete mapping at the conclusion of each influenza season. In short, we scaled unvaccinated susceptible compartment values to ensure the updated age-level proportion was matched. For a full description, see Section 2.3.3 of the S1 Text.

For use in our observation model, we tracked the incidence $Z_i^m(y)$ of new strain $m$ influenza infections for those aged $i$ in influenza season $y$ as a rate per 100,000 population:

$$Z_i^m(y) = \left( \int_{y-1}^{y} \gamma_{1,m}(E_i^{N,m} + E_i^{V,m}) \, \mathrm{dt} \right) \times 100{,}000. \qquad (2)$$

We highlight at this point that, when constructing mathematical models to capture seasonal influenza transmission dynamics, it is commonplace to consider a risk-stratified population, divided into individuals at low or high risk of complications associated with influenza. However, given risk group status most profoundly affects health outcome condition on infection status, to lessen model complexity we assumed epidemiological processes were not dependent on risk status. Given those dependencies, for our model fitting purposes we ran the model, as described above, with a population that was not stratified by risk status. In effect,

all data inputs and model outputs were averaged across risk groups. Next, to carry out health economic evaluations, we recovered the risk-group specific epidemiological outcomes by running the model with the relevant age distribution and vaccine uptake data (specific to a given risk group), whilst feeding in the population level (risk group averaged) force of infection.

**Observation model.** The observational model segment of the mathematical framework linked the GP consultation data with the number of infections due to circulating influenza viruses in the population. Our interest resided in the number of ascertainable cases, where we assumed that each individual infected by the strain of influenza under consideration had a probability of being ascertainable. It should be noted that we considered ascertainable cases to encompass symptomatically infected individuals going to the GP, being recorded as having ILI, and having a detectable influenza viral load.

Ultimately, we wanted to obtain model estimates of the proportion of individuals aged $i$ experiencing cases of ascertainable influenza (of strain type $m$) in influenza season $y$ through scaling the incidence $Z_i^m(y)$.

Yet, it may be unwarranted to assume the ascertainment probability to be consistent across influenza seasons and different ages. As a consequence, we employed season-specific ascertainment functions with flexibility for ascertainment probabilities to be modulated by age. To that end, we enforced ascertainment probability profiles to be piecewise linear. We established the desired behaviour through two scaling factors.

The first type of ascertainment parameter was an absolute value for ascertainment of an influenza case in an 100+ aged individual, obtained per influenza season $y$ (denoted $\epsilon_y$). We highlight at this juncture the inherent assumption that ages 100+ have the highest ascertainment probability.

The second type of ascertainment parameter was a relative scaling (between 0 and 1) at five knot points, ages where the pieces of the linear piecewise function met (the selected ages were 0, 2, 18, 65, 85). We kept age-dependent ascertainment probability proportionality factors, labelled $\tilde{\epsilon}_i$, consistent across all influenza seasons.

Combining the two scaling factors, the ascertainable influenza cases in season $y$, $Z_m^+(y)$, obeys:

$$Z_m^+(y) = \tilde{\epsilon}_i \epsilon_y Z_i^m(y). \tag{3}$$

We inferred all ascertainment-related parameters ($\epsilon_y$, $\tilde{\epsilon}_i$) by comparing the total predicted incidence in a given year to the calculated GP consultation rate (Eq (1)).

**Parameter fitting.** To realise a model capable of generating influenza-attributed ILI GP consultations estimates that resemble the empirical data (Eq (1)), we sought to fit the following collection of model parameters (Table 2): transmissibility of each influenza virus strain ($\beta_m$), modified susceptibility to strain $m$ given infection by a strain $m$ type virus the previous influenza season ($a$), carry over cross-reactivity protection between influenza B lineages ($b$), residual protection carried over from the prior season influenza vaccine ($\xi$), a collection of four age-specific susceptibilities ($\sigma_{0-17}$, $\sigma_{18-65}$, $\sigma_{65-84}$, $\sigma_{85+}$), an ascertainment probability per influenza season for those aged 100+ ($\epsilon_y$), and relative ascertainment values at five designated knot points (ages 0, 2, 18, 65, 85, respectively) for constructing a piecewise linear ascertainment function with respect to age ($\tilde{\epsilon}_{0\text{yrs}}$, $\tilde{\epsilon}_{2\text{yrs}}$, $\tilde{\epsilon}_{18\text{yrs}}$, $\tilde{\epsilon}_{65\text{yrs}}$, $\tilde{\epsilon}_{85\text{yrs}}$).

All the parameters listed above were used directly in the model, with the exception of the basic reproduction number for each influenza virus strain, $R_{0_m}$, which corresponded to the dominant eigenvalue of the next generation matrix for that particular strain [43]. Instead, on a given iteration of the inference scheme we used $R_{0_m}$ to calculate the transmissibility of each

influenza virus strain, $\beta_m$. Note that the computed value of $\beta_m$, when used in combination with the recovery rate $\gamma_2$ and the age-structured contact array, would generate a next generation matrix with $R_{0_m}$ as its dominant eigenvalue.

Simulations ran from the 2009/10 influenza season up to and including the 2017/18 influenza season and ran for a specified number of seasons with one parameter set. We fit to the 2012/13 influenza season onward, which omitted the time period containing influenza seasons that had no influenza B lineage typing data (specifically 2009/10 and 2011/12, Fig 1).

The summary statistics for our fitting procedure were similar to those used when fitting the non-age structured variant of the model to the available data [16]. In detail, the first component was a within-season temporal profile check; the peak in influenza infection (combining those in latent and infectious states) could not occur after February for any age class in any influenza season. The second component defined the metric to measure the correspondence of the model predicted influenza-attributed ILI GP consultations versus the observed data. We worked with a metric akin to Poisson deviance. Accounting for influenza season $y$, age $i$, and strain stratification $m$, we defined our deviance measure to be

$$\mathrm{DEV} = 2 \sum_y \sum_i \sum_m \left( C_{m,i,y} \ln \left( \frac{C_{m,i,y}}{M_{m,i,y}} \right) - (C_{m,i,y} - M_{m,i,y}) \right), \tag{4}$$

with $C_{m,i,y}$ the observed value for strain $m$ amongst those aged $i$ in influenza season $y$, and $M_{m,i,y}$ the model estimate for strain $m$ amongst those aged $i$ in influenza season $y$.

We carried out parameter optimisation using the OPTIM package for JULIA [44]. By carrying out 160 runs of an adaptive particle swarm algorithm [45] (for 10,000 iterations apiece), we amassed 100 particles conforming to the temporal criteria and returning a deviance measure (given by Eq (4)) below 30,000.

## Simulating vaccination strategies

Post employing the inference scheme and acquiring parameter distributions representing those values giving the closest correspondence to the empirical evidence, given the stated model assumptions, in the next phase of the study our attention turned to performing a health economic appraisal of conjectured vaccination programmes in England.

To enable comparison between counterfactual vaccination scenarios, we needed to simulate each vaccine programme of interest. This comprised two steps. First, we parameterised the model by sampling parameter values from the parameter distributions obtained by our parameter fitting scheme. Second, we ran the model with the counterfactual vaccine scenario for that influenza year and scenario applied, thus generating the estimated number of infections that would have occurred under that alternative vaccination strategy.

All model simulations began at the start of the 2009/2010 influenza season (i.e. September 2009), though the time horizon applicable for assessing cost-effectiveness began from the 2012/13 influenza season (further details are provided in the "Cost-effectiveness analysis" subsection of the Methods). For our baseline strategy, the 2009/10 influenza season matched historical vaccine uptakes and efficacies of the pandemic influenza vaccine (for both low-risk and at-risk groups), while for the 2010/11 influenza seasons onward only vaccination of at-risk groups (matching historical uptake and efficacies) were simulated.

The vaccination strategies we simulated may be batched into three types, with the purpose of offering insights into three distinct questions. In batch one, we aimed to determine how the allowable vaccine price dose for introducing vaccination to low-risk groups (relative to a baseline strategy maintaining the historical vaccination coverage of at-risk individuals) was

impacted by the breadth of coverage amongst children, adolescents and the elderly. In batch two, we extended our investigation to identify the extent of vaccination coverage amongst low-risk youth and elder age classes that would maximise cost-effectiveness, while maintaining the historical vaccination coverage of at-risk individuals. Finally, in batch three we sought to identify whether incremental expansion in the covered low-risk vaccination age range would be judged as cost-effective. The specifics of implementing each batch are elaborated upon below.

**Paediatric and elder-age centric campaigns.** To begin, we simulated vaccination strategies that paired together child, adolescent and and elder-age based focused components of varying coverage. Our intent was to determine how the allowable vaccine price dose for introducing vaccination to low-risk groups (relative to a baseline strategy maintaining the historical vaccination coverage of at-risk individuals) was impacted by the breadth of coverage amongst children, adolescents and the elderly.

For paediatric campaigns, we were interested in coverage that either did or did not include pre-school, primary school and secondary school aged children. We therefore considered six paediatric campaigns with the following age coverage: None, 4-10yrs (primary school only), 4-16yrs (primary and secondary school), 2-4yrs (pre-school only), 2-10yrs (pre-school and primary school aged children), and 2-16yrs (preschool, primary and secondary school). Per paediatric strategy, we tested five elder-age centric programmes: 50 years and above, 60 years and above, 70 years and above, 80 years and above, and 90 years and above.

Pairing together each of the paediatric and elder-age centric strategies resulted in 30 differing coverage combinations. For each of these strategies, we made age-based assumptions with regards to vaccine uptake. For the age groups covered by the paediatric campaign, we set vaccine uptake at 60% (amongst both low-risk and at-risk groups). For the age groups covered by the elder-age centric programme, vaccine uptake matched the historical data (i.e. vaccine uptake for those aged 65 years and above was set to equal the observed vaccine uptake data per influenza season). For those age groups not covered by either a paediatric or elder-age centric strategy, vaccine uptake was 0% for the low-risk group, whilst for the at-risk group vaccine uptake was as per the observed data within each respective influenza season.

**Sweep of age banded vaccine programmes.** For the next batch of vaccination strategy simulations, we sought to identify the extent of vaccination coverage amongst low-risk youth and elder age classes, while maintaining the historical vaccination coverage of at-risk individuals, that would maximise cost-effectiveness. Switching perspective, we also determined the optimal level of age coverage for a fixed cost of seasonal influenza vaccine (i.e. the vaccine age coverage that minimised overall cost). Optimal age ranges were calculated at £1 increments for cost of an influenza vaccine.

We considered the joint inclusion of young-age centric and elder-age centric vaccination programmes. The studied age ranges for the young-age centric strategies were $2 - \hat{T}$, $\hat{T} \in \{2, 4, \ldots, 98, 100+\}$, and in a similar manner for the elder-age centric strategies, $\tilde{T} - 100+$, $\tilde{T} \in \{2, 4, \ldots, 98, 100+\}$. The lower bound for the young-age centric component was set at two years of age to correspond with the youngest age included within the current (at the time of writing) paediatric vaccination programme in England. We omitted paired strategies with overlapping age ranges (where $\hat{T} > \tilde{T}$). In addition, we analysed the sole addition (alongside vaccination of those at-risk) of either a young-age centric low-risk or an elder-age centric low-risk vaccination campaign.

These arrangements gave a total of 1,375 vaccination schemes. For each of these schemes, we again applied age-based assumptions with regards to vaccine uptake. For ages spanned by the young-age centric component, we set vaccine uptake at 60% amongst both low-risk and at-risk groups. For the age groups covered by the elder-age centric component, vaccine uptake

matched the historical data (i.e. vaccine uptake for those aged 65 years and above was set to equal the observed vaccine uptake data per influenza season). For those age groups not covered by either a young-age centric or elder-age centric component, vaccine uptake was 0% for the low-risk group, whilst for the at-risk group vaccine uptake was as per the observed data within each respective influenza season.

**Incremental expansion of low-risk vaccination programmes.** For our final set of vaccination strategy scenario simulations, we took a differing baseline perspective. Our question hinged on whether an incremental adjustment or 'step change' in the covered low-risk vaccination age range would be judged as cost-effective. In other words, determining whether the inclusion of a couple of additional single year age groups would produce a positive threshold vaccine dose price (post deduction of administration charges).

Under this alternative setting, the incremental expansion in low-risk vaccination programme worked in the follow manner. As an example, given a reference strategy with coverage of those aged 2–10yrs and 70–100+yrs, an incremental expansion of the young-age centric programme corresponded to the revised vaccine strategy covering those aged 2–12yrs (upper bound of age coverage increased by two years, from 10 year olds to 12 year olds) and 70–100 +yrs (elder-age component unmodified). Furthermore, if the incremental expansion was instead implemented for the elder-age centric programme, the revised age coverage of the vaccination strategy would be 2–10yrs (young-age centric component unmodified) and 68–100 +yrs (the lower bound of age coverage reduced by two years, from 70 year olds to 68 year olds).

Generalising, with a reference strategy covering the ages $2 - \hat{T}$yrs and $\tilde{T} - 100$+yrs (with $\hat{T} < \tilde{T}$), an incremental expansion of the young-age centric programme corresponded to a revised age coverage of $2 - (\hat{T} + 2)$yrs and $\tilde{T} - 100$+yrs, whilst incremental expansion of the elder-age centric programme covered ages $2 - \hat{T}$yrs and $(\tilde{T} - 2) - 100$+yrs.

## Economic model

The economic model mapped outputs from the epidemiological model to clinical outcomes (specifically, GP consultations, hospitalisation and mortality events). The healthcare costs and quality-adjusted life years (QALYs) attributable to each vaccination strategy were compared to a baseline strategy and the incremental cost-effectiveness ratio of each strategy was estimated. We then generated the threshold vaccine dose price; that is, the maximum amount the healthcare system is willing to pay per vaccine dose, given the associated health benefits. Positive prices per vaccine dose below this threshold price will tend to generate positive net health benefits, whilst negative prices per vaccine dose offer no incentive to the manufacturer to provide the vaccine.

The economic evaluation was conducted from a United Kingdom National Health Service (NHS) and personal social services perspective with costs valued in pounds sterling (2016-17 prices). The following sections outline the clinical, cost and health utility parameters that fed into our economic model.

**Clinical parameters.** With regards to severity, influenza illness can lead to a range of clinical outcomes (asymptomatic, symptomatic (no primary healthcare consultation), GP consultation, non-fatal hospitalisation, death).

We took from the literature estimates of the per-infection probability of developing clinical symptoms [46], and a relative risk of consulting a GP if in the at-risk group and infected with influenza [47] (see Table 3). Though our model fit provided a (risk group averaged) per-infection probability of consulting a GP, we could recover the risk group specific ascertainment rates (further details are given in Section 3.1 of the S1 Text).

**Table 3. Symptomatic case and GP consultation parameters.**

| Parameter | Estimate | Uncertainty | Source |
|---|---|---|---|
| Symptomatic probability | 0.406 | Triangular on [0.309-0.513] | [46] |
| Relative risk of consulting a GP (at-risk group) | 1.51 | Normal($\mu = 1.51$, sd = 0.18) | [47] |

Data on hospitalisation-related events (outpatient attendances, non-fatal inpatient admission, fatal illness) were provided through Hospital Episode Statistics (HES) records, spanning September 2012—August 2017. HES is a database containing individual-level records of all admissions, Accident and Emergency attendances and outpatient appointments at NHS hospitals in England. For our analysis, we used aggregated data corresponding to patient records that included a seasonal influenza ICD-10 diagnostic code (J10 or J11 code) in any diagnosis field. These data were additionally subdivided into at-risk and non-risk groups based on a collection of risk group related ICD-10 codes (Table B in S1 Text). Data extract specifics are expanded upon in Section 3.2 of the S1 Text.

For each hospitalisation-related event and age grouping, we determined the relative likelihood of event occurrence compared to a single GP consultation. Due to the HES records having no associated influenza type information (i.e. stating whether a hospitalisation case was due to influenza A or influenza B), and to allow inclusion of heterogeneities in case severity between influenza types, we implemented a set of assumptions regarding permissibility of event occurrence dependent upon age that were discerned from prior statistical modelling studies. As a result, we generated differing hospitalisation case rates for influenza A and influenza B, stratified into 12 age bands (0–1yrs, 2–5yrs, 6–9yrs, 10–19yrs, 20–29yrs, 30–39yrs, 40-49yrs, 50–59yrs, 60–64yrs, 65–74yrs, 75–84yrs, 85+yrs), with uncertainty distributions procured via bootstrapping (see Section 3.3 of the S1 Text).

Quantifying the burden of mortality resulting from influenza is not straightforward due to under-reporting as a cause of death. Consequently, we gleaned from the literature estimates of the fraction of influenza deaths that occur in the hospital so we may appropriately scale HES-informed mortality rates to an approximate, all-cause influenza mortality measure [48] (Table 4).

**Costs.** In pursuing attainment of a well-informed, evidence-based economic model, we sought to gather relevant economic cost estimates from recent standard and/or published data sources.

The unit cost of a GP consultation was provided by Curtis & Burns [49], with a consultation cost estimate of £37.40, and a lognormal uncertainty distribution (mean = 37.40, standard deviation = 8.4).

We synthesised age- and risk-stratified hospitalisation case costs from the provided HES records (see Section 3.2 of the S1 Text), with average costs attained over the five year time-frame spanned by the data (September 2012–August 2017 inclusive). In brief, the supplied cost estimates (based on HES variables) were generated combining data on primary diagnosis

**Table 4. Proportion of mortality events occurring in- and out-of-hospital (informed using data from Table 3 of Matias et al. [48]).**

| Age group (yrs) | % of deaths in hospital | % of deaths out-of-hospital |
|---|---|---|
| 0–49 | 100% | 0% |
| 50–64 | 75% | 25% |
| 65–74 | 65% | 35% |
| 75+ | 50% | 50% |

**Table 5. Non-fatal case decrements in health utilities.**

| Parameter | Estimate | Uncertainty | Source |
|---|---|---|---|
| Asymptomatic case (All ages) | 0.00 | — | — |
| Outpatient event (All ages)† | 0.00 | — | — |
| Non-hospitalised, symptomatic case (0–16 yrs) | 0.00749 | Normal($\mu$ = 0.00749, sd = 0.00085) | [50, 51] |
| Non-hospitalised, symptomatic case (17+ yrs) | 0.00820 | Normal($\mu$ = 0.0082, sd = 0.0018) | [50, 51] |
| Non-fatal, hospitalised case (0–16 yrs) | 0.016 | Normal($\mu$ = 0.016, sd = 0.0018) | [52, 53] |
| Non-fatal, hospitalised case (17+ yrs) | 0.018 | Normal($\mu$ = 0.018, sd = 0.0018) | [52, 53] |

† Outpatient events did not carry any QALY loss. Outpatient attendances occured subsequent to a previous hospital admission and therefore the QALY loss associated with the prior hospital admission has already been accounted for.

ICD-10 codes, secondary diagnoses ICD-10 codes, procedures carried out and length of stay in hospital. We then formed influenza type specific costs (Section 3.4 of S1 Text), adhering to our previously discussed event occurrence assumptions (Section 3.3 of S1 Text). There was assumed to be no cost associated with an influenza attributed out-of-hospital mortality event.

For the baseline analysis, we assumed the cost of vaccination administration to be £10 per dose (Department of Health, personal communication).

**Health utilities.** Health utility decrements associated with non-fatal cases of influenza were extracted from the literature [50–53] (see Table 5). QALY loss due to death was commensurate with the remaining number of expected healthy years of life remaining (details of the calculation are given in Section 3.5 of the S1 Text).

**Cost-effectiveness analysis.** The process to evaluate cost-effectiveness had similar characteristics to that of approaches used by Tsuzuki *et al.* [9] and Baguelin *et al.* [11]. It involved using the reconstructed epidemic profiles under the actual (historically undertaken) vaccination programme. We then used the same model to estimate the number of infections for each influenza strain by age and risk group that would have occurred under an alternative vaccination programme. In the final step, the economic model mapped the epidemiological outcomes to clinical outcomes, providing monetary costs and health utility quantities per strategy that could be used to judge whether a proposed strategy was cost-effective compared to a baseline strategy.

For clarity, there was a distinction between the time period covered by a single model simulation and the time horizon used to assess strategy cost-effectiveness. For initialising the system, all model simulations began from the 2009/10 influenza season. For our cost-effectiveness assessment, the time horizon of the base case model spanned the 2012/13 to 2017/18 influenza seasons, reflecting the historical influenza season data the model had been fit too.

To comply with the JCVI's guidelines, we considered two criteria that an alternative strategy needed to satisfy to be judged as cost-effective. Firstly, that for the most likely set of parameters the mean discounted costs and outcomes should be evaluated against a £20,000 cost-effectiveness threshold value for a QALY [54]. Secondly, to account for uncertainty, 90% of all posterior parameters should generate cost-effective results at a threshold of £30,000 per QALY.

When performing analysis for the second criterion, we conducted a probabilistic sensitivity analysis for each of the alternative intervention strategies. Explicitly, we sought the threshold dose prices at the 10th percentile of simulated values (with the cost-effectiveness threshold set at £30,000 per QALY). For each vaccination scheme, we performed ten transmission model simulations (with all strategies using the same set of ten epidemiological particles). To generate 100 Monte Carlo parameter samples, we first drew epidemiological outcomes from outputs generated from one of the ten epidemiological parameter particle sets. We then sampled health economic parameters from appropriate distributions to produce the desired health economic outputs.

In primary analyses, we used a discounting rate of 3.5% per year (for both healthcare costs and QALYs). As an alternative scenario, we also evaluated the effects of applying a 1.5% discount rate to both monetary costs and health impacts. Furthermore, in all analyses we presumed that 10% of vaccines ordered were not administered ('vaccine wastage').

## Simulation and software specifics

Model fit and simulations were performed in JULIA v1.0–v1.2. The system of ODEs were solved numerically with the package DIFFERENTIALEQUATIONS v6.6.0, and parameter optimisation was carried out using the package OPTIM v0.19.2. Data processing and production of figures were carried out in MATLAB R2019b.

# Results

## Heightened propagation of natural immunity over vaccine-induced immunity

We invoked the particle swarm optimisation algorithm, accumulating 100 parameter sets to determine credible values of the model parameters (Table 6 and Fig 2). Samples were obtained

**Table 6. Values for the basic reproduction number, exposure history, susceptibility and ascertainment probability parameters inferred fitting to the empirical data.** Estimates computed from 100 particles. Numbers inside brackets indicate 95% credible intervals. All values are given to 4 d.p.

| Description | Notation | Median [95% credible interval] |
|---|---|---|
| **Transmission parameters** | | |
| A(H1N1)pdm09 | $R_{0_{A(H1N1)pdm09}}$ | 2.0544 [1.8001, 2.2127] |
| A(H3N2) | $R_{0_{A(H3N2)}}$ | 2.0987 [1.8607, 2.3007] |
| B/Victoria | $R_{0_{B/Victoria}}$ | 1.8709 [1.6567, 2.0246] |
| B/Yamagata | $R_{0_{B/Yamagata}}$ | 1.9687 [1.7392, 2.1340] |
| **Exposure history parameters** | | |
| Natural infection in prior influenza season | $a$ | 0.7133 [0.6757, 0.7353] |
| Type B influenza cross-reactivity | $b$ | 1.0000 [1.0000, 1.0000] |
| Prior influenza season vaccine efficacy propagation | $\xi$ | 0.0000 [0.0000, 0.0260] |
| **Susceptibility parameters** | | |
| Ages 0–17 | $\sigma_{0-17}$ | 0.7148 [0.6375, 0.8393] |
| Ages 18–64 | $\sigma_{18-64}$ | 0.6277 [0.5288, 0.7387] |
| Ages 65–84 | $\sigma_{65-84}$ | 0.7766 [0.6201, 0.8664] |
| Ages 85+ | $\sigma_{85+}$ | 1.0000 [0.6769, 1.0000] |
| **Reference ascertainments (ages 100+)** | | |
| 2012/13 | $\epsilon_{2012/13}$ | 0.0060 [0.0036, 0.0125] |
| 2013/14 | $\epsilon_{2013/14}$ | 0.0039 [0.0023, 0.0089] |
| 2014/15 | $\epsilon_{2014/15}$ | 0.0096 [0.0061, 0.0204] |
| 2015/16 | $\epsilon_{2015/16}$ | 0.0134 [0.0080, 0.0312] |
| 2016/17 | $\epsilon_{2016/17}$ | 0.0065 [0.0040, 0.0148] |
| 2017/18 | $\epsilon_{2017/18}$ | 0.0252 [0.0150, 0.0600] |
| **Relative ascertainment factors** | | |
| 0 year old | $\tilde{\epsilon}_{0yrs}$ | 0.2087 [0.1119, 0.3368] |
| 2 year old | $\tilde{\epsilon}_{2yrs}$ | 0.1383 [0.0599, 0.2404] |
| 18 year old | $\tilde{\epsilon}_{18yrs}$ | 0.2367 [0.1263, 0.3903] |
| 65 year old | $\tilde{\epsilon}_{65yrs}$ | 0.6974 [0.3868, 1.0000] |
| 85 year old | $\tilde{\epsilon}_{85yrs}$ | 0.5116 [0.3449, 0.9297] |

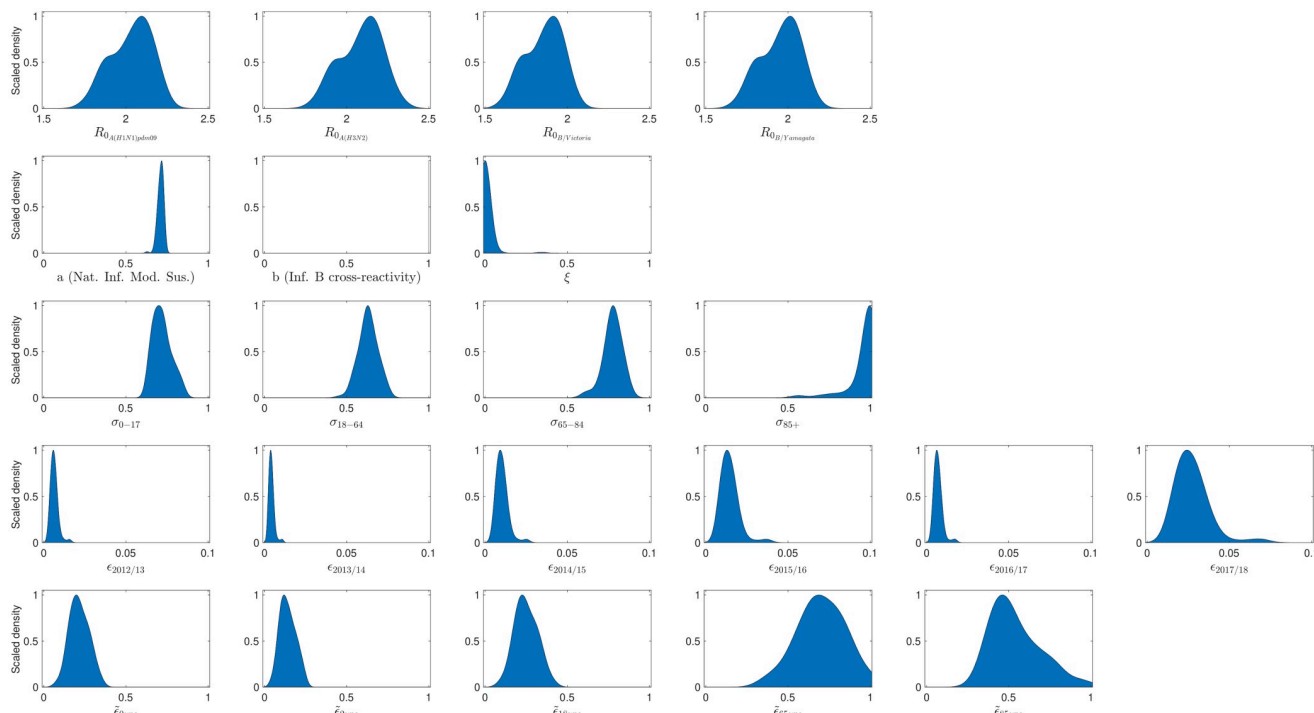

**Fig 2. Acquired parameter distributions from the optimisation scheme, fitting to the empirical data.** Parameter densities generated from 100 particles that each returned an error below 30,000. Blue shaded areas outline the scaled density. (**Row one**) Basic reproduction numbers by strain. (**Row two**) Immunity propagation parameters. (**Row three**) Susceptibility by age group. (**Row four**) Ascertainment probabilities for those aged 100+, by influenza season. (**Row five**) Relative ascertainment by age. Particularly noteworthy outcomes include: prior season influenza B cross-reactivity and vaccine carry over had little impact on present influenza season susceptibility; the highest ascertainment probability for the 100+ age group occurred in the 2017/18 influenza season; amongst the knot ages to produce the piecewise linear ascertainment function, the greatest ascertainment was observed for those 65 years of age.

after 10,000 iterations of the algorithm were completed. All 100 retained parameter sets adhered to the mandatory temporal criteria of peak infection in each influenza season occurring prior to March, and each returned a deviance measure below 30,000 (Eq (4)).

Viewing the attained distributions for $R_0$ per strain, the estimates for the two influenza A subtypes are similar; both are larger than the corresponding estimates for the two type B lineages, with B/Yamagata transmissibility estimates exceeding those for B/Victoria.

Amongst the three exposure history parameters ($a$, $b$, $\xi$), the parameter seemingly imparting impact on the transmission dynamics was the modification to susceptibility due to natural infection in the previous influenza season ($a$). If infected by a given strain last influenza season, the inferred weighted median estimate of 0.7159 corresponds to an approximate 28% reduction in susceptibility to the current influenza season variant of that strain type. We see little support for carry over of prior influenza season vaccine efficacy ($\xi$), with the majority of the inferred distribution massed near 0. Finally, with all retained particles having $b = 1$, conferred influenza B cross-reactive immunity was not retained into the subsequent influenza season.

Comparing between the age susceptibility parameters, the greatest susceptibility was evidently found to be amongst the 85+ age category. Further, the child and adolescent category had a near 10% reduction in susceptibility relative to those aged 65–84 (0.7121 vs 0.7779). We also found a similar (relative) percentage decrement between adults (18–65 years) and young people (0–17 years).

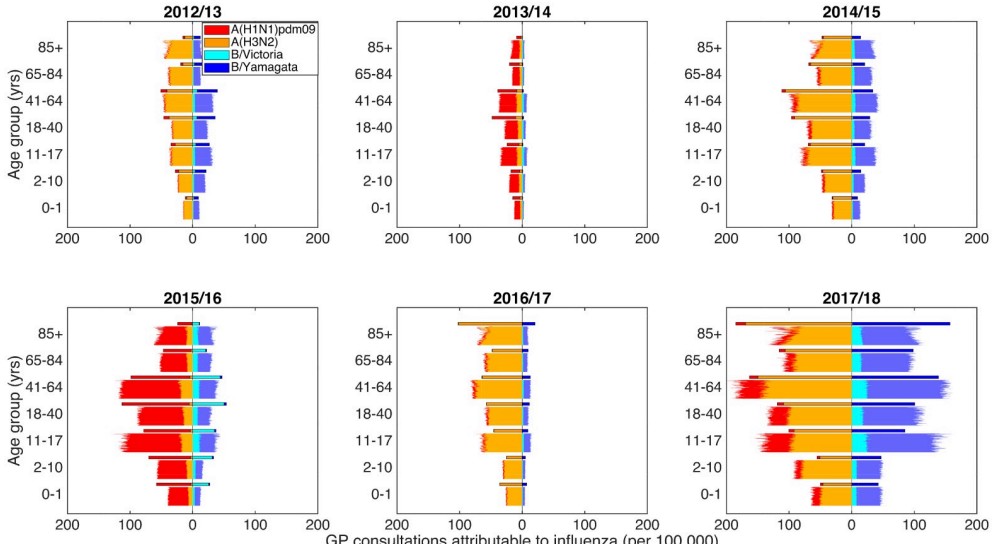

**Fig 3. Predictive distributions for influenza positive GP consultations per 100,000 population.** Stratified by influenza season and age bracket, we present back-to-back stacked bars per simulation replicate, with 100 replicates performed, each using a distinct parameter set representing a sample from the parameter distributions obtained from our parameter fitting scheme. Each panel corresponds to an individual influenza season. Within each panel, all age brackets are topped out by a thicker stacked horizontal bar plot, corresponding to the strain-stratified point estimates for the empirical data. In all panels, the left side depicts data pertaining to type A influenza viruses (red shading denoting the A(H1N1)pdm09 subtype, orange shading the A(H3N2) subtype). In an equivalent manner, the right side stacked horizontal bars present similar data for type B influenza (cyan shading denoting the B/Victoria lineage, dark blue shading the B/Yamagata lineage). We see a reasonable qualitative model fit to the data.

Inferred ascertainment probabilities for those aged 100+ ($\epsilon$), across all influenza seasons, ranged up to 0.06. The highest ascertainment probabilities were found for the recent 2017/18 influenza season. Comparing across ages, there was minor variation in ascertainment probability for those aged 0–18, with a peak at 65 years of age, a small decline up to 85 year of age, followed by increases as age rised towards 100+ (Fig. E in S1 Text).

To check the goodness-of-fit between our model and the available data, we performed 100 independent simulations using the parameter sets acquired from the optimisation fitting procedure. Therefore, although the model is deterministic, we generated variability in epidemic composition due to the inferred distribution for the underlying parameters. The incidence of each of the four influenza types over each influenza season and age group was converted to attributed GP consultations (per 100,000). We generally obtained a reasonable fit to the data across age ranges and strains, generally mimicking the pattern of strain composition and similar magnitudes of consultation load in each influenza season (Fig 3). There was also alignment between the empirical data and model averaged forecast results, showing the model was capable of predicting the general pattern of high and low observations (Fig. F in S1 Text).

Considering type A influenza, we captured the empirical pattern of alternating dominance by A(H3N2) and A(H1N1)pdm09 subtypes for the years 2012/13 to 2016/17, followed by A(H3N2) dominating again in 2017/18. For influenza B, we obtained modest agreement for the overall magnitude of GP consultations as a result of type B, but this was not always in complete agreement with the subtype composition.

Switching focus to age groups, the simulated severity of consultation load was typically similar to that of the empirical data. One notable exception was within the eldest age grouping

(85+ years), where we found less favourable agreement in the latter three influenza seasons (2015/16 through to 2017/18). Specifically, the simulated amount of total GP consultations attributable to influenza was overestimated for the 2015/16, and underestimated in both 2016/17 and 2017/18.

## Simulating vaccination programmes

**Increase in threshold vaccine dosage prices for vaccine programmes including a paediatric campaign.** Relative to a reference strategy of at-risk group vaccination, we studied 30 combinations of paediatric and elder-age centric programmes that extended vaccine uptake to low-risk individuals within select age groupings. Recall that our 30 age coverage combinations paired one of six paediatric campaigns (None, 4-10yrs, 4-16yrs, 2-4yrs, 2-10yrs, and 2-16yrs), with one of five elder-age centric programmes (50 years and above, 60 years and above, 70 years and above, 80 years and above, 90 years and above).

For each strategy we computed a threshold price per additional vaccine dose (threshold vaccine dose price), reflecting the value at which the strategy becomes cost-effective (relative to the reference strategy) at a WTP of £20,000 per QALY, with 3.5% discounting for monetary costs and health effects. We simulated each vaccine scheme with the most likely set of parameter values; for epidemiological parameters we used the set of parameter values that returned the lowest error, and for health economic parameters we took the average estimate.

In the absence of a paediatric component to the low-risk vaccination programme (those with a paediatric campaign coverage of "None"), the elder-age strategy returning the lowest threshold vaccine dose price corresponded to low-risk vaccination being provided to those aged 70 and above (threshold vaccine dose price of £1.63). Either contracting or expanding the age coverage resulted in a larger threshold price per vaccine dose (£8.02 for a programme targeting those aged 90 and above, £8.59 for a programme targeting those aged 50 and above).

Adding any of our considered forms of paediatric low-risk vaccination campaign led to a rise in the threshold vaccine dose price (Fig 4). The programme producing the highest threshold vaccine dose price, of £29.12, was a coverage combination of those primary school aged together with those aged 90 years and above. In addition, across all tested paediatric campaigns, a lesser breadth of elder-age coverage resulted in a higher threshold vaccine dose price. For example, under the primary school aged coverage strategy (4-10yrs), moving from elder-age coverage encompassing those aged 90 years and above to those aged 50 years and above saw a drop in threshold vaccine dose price from £29.12 to £10.54. Such decreases in threshold vaccine dose price were less stark for paediatric coverage strategies covering more age groups (2-16yrs), with prices ranging from £21.91 (for elder-age coverage spanning those aged 90 years and above) to £12.55 (with coverage including those aged 50 years and above).

Finally, we observed that paediatric campaigns including solely primary or both primary and secondary age groups returned higher threshold vaccine dose prices compared to those campaigns that also included pre-school coverage. This relationship was independent of the level of coverage amongst the elder age classes.

**Identification of preferred pairings of young-age centric and elder-age centric vaccine programmes.** Relative to a reference strategy of at-risk group vaccination, we studied 1,375 alternative vaccination programmes that each administered vaccines to a differing proportion of the low-risk population (in addition to vaccination of those at-risk). Each of the tested alternative programmes spanned a differing range of age classes. We began by studying paired interventions combining our young-age centric and elder-age centric vaccination strategies. Recall that we omitted paired strategies with overlapping age ranges. We also considered the

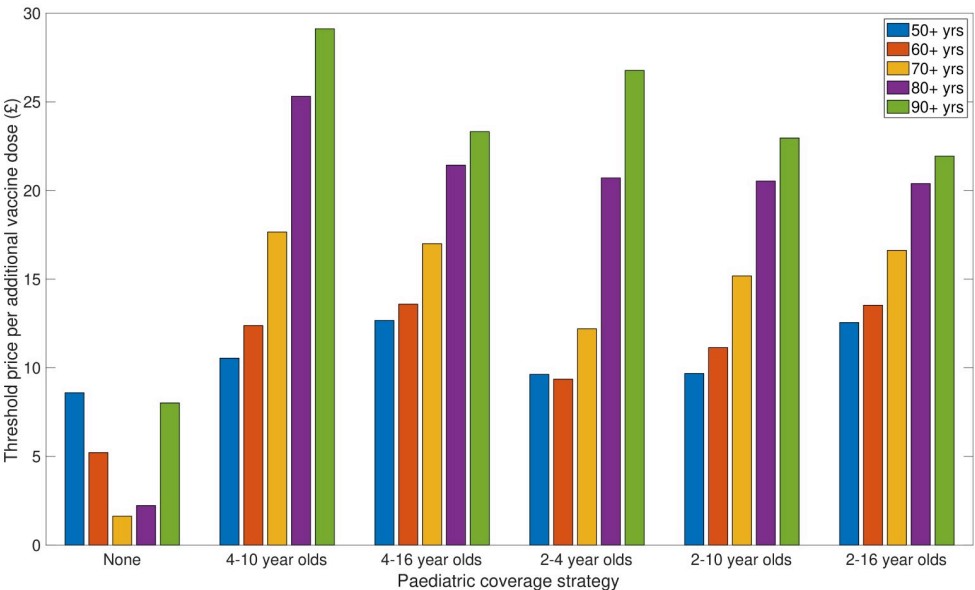

**Fig 4. Threshold price per additional vaccine dose enabling low-risk uptake amongst paediatric and elder-age centric age groups to be deemed cost-effective.** Threshold vaccine dose prices reflect the value at which the programme becomes cost-effectiveness at a WTP of £20,000 per QALY, with 3.5% discounting for monetary costs and health effects. In all considered strategies, we assumed 10% vaccine wastage and an administration fee per deployed vaccine of £10. In order of plotting group, the paediatric strategies covered the following age ranges: None, 4-10yrs (primary school only), 4-16yrs (primary and secondary school), 2-4yrs (pre-school only), 2-10yrs (pre-school and primary school aged children), and 2-16yrs (preschool, primary and secondary school). Per paediatric strategy, we considered five elder-age centric programmes, each having a varying breadth of age coverage: 50 years and above (blue bars), 60 years and above (orange bars), 70 years and above (yellow bars), 80 years and above (purple bars), 90 years and above (green bars). For full results, see S1 File.

sole addition (alongside vaccination of those at-risk) of either a young-age centric low-risk campaign or a elder-age centric low-risk campaign.

Initially, we evaluated cost-effectiveness against a £20,000 threshold value for a QALY, with each vaccine scheme simulated with the most likely set of parameter values. In short, all programmes were cost-effective, returning a positive threshold dose price per vaccine (post deduction of the administration charge). Full results are provided in Fig. G of the S1 Text and S2 File.

We explore in greater detail the cost-effectiveness analysis accounting for uncertainty, where values were set where 90% of simulations generated cost-effective results at a willingness-to-pay threshold of £30,000 per QALY (Fig 5). A full breakdown of these threshold vaccine dose prices are also given in S2 File.

Inspecting paired strategies, across the breadth of young-age centric programme component converage levels there was striking banding, with particular peaks in dose price amongst campaigns covering primary school children (2–4yrs to 2–8yrs) and then again when coverage expanded to include young adults (2–18yrs to 2–30yrs). Fixing the coverage of the young-age centric programme at a given age, the optimum elder-age centric programme (in terms of maximising the threshold vaccine dose price) tended to contain only the eldest ages (generally 80 years and above). We found as age coverage broadened for elder-age centric strategies, vaccine dose price thresholds generally lowered.

The optimum paired strategy, with a threshold vaccine dose price of £37.60, covered those 2–6 years of age in conjunction with those 98+ years of age. The paired strategy returning the

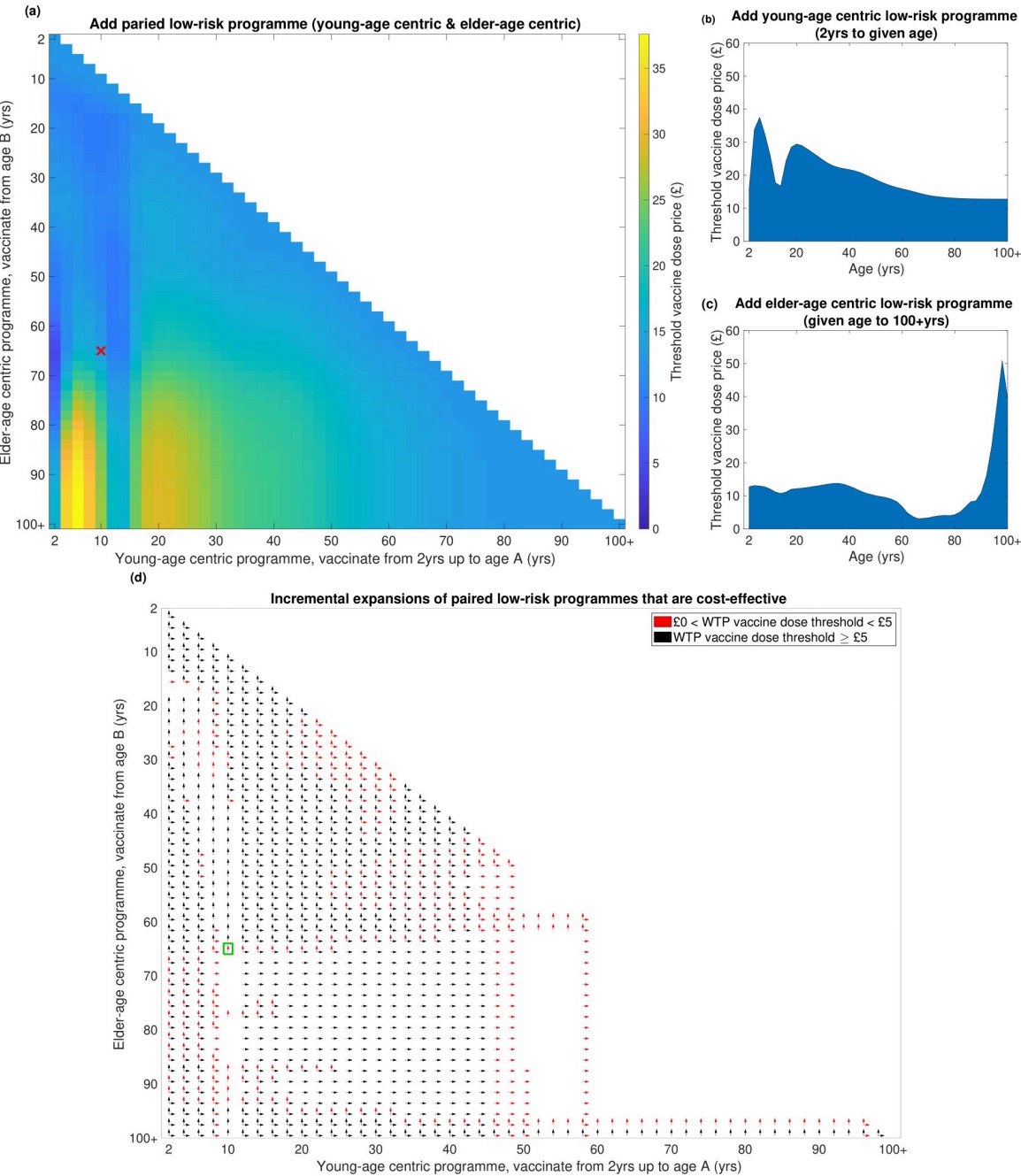

**Fig 5. Threshold price per additional vaccine dose for additional vaccines required to add a low-risk vaccine programme (relative to an at-risk only vaccination programme).** Threshold vaccine dose prices reflect the cost at which 90% of all simulations generate cost-effective results at a WTP of £30,000 per QALY, with 3.5% discounting for monetary costs and health effects. In all strategies we assumed 10% vaccine wastage and an administration fee per deployed vaccine of £10. **(a)** Paired low-risk programmes (combined young-age centric and elder-age centric coverage). Shading transitioning from dark shading to light shading depicts shifts from low to high prices. The white region corresponds to omitted strategies, where young-age centric and elder-age centric coverage would overlap. The red cross marks the coverage offered by the current seasonal influenza vaccine policy for England. **(b)** Addition of a low-risk young-age centric programme only. **(c)** Addition of a low-risk elder-age centric programme only. **(d)** Cost-effectiveness assessments of incrementally expanding paired low-risk vaccine programmes. Vertical arrows denote incremental expansions of the elder-age centric component (e.g. going from 70-100+yrs to 68-100+yrs) of a paired vaccination programme that were evaluated as cost-effective. Similarly, horizontal arrows denote incremental expansions of the young-age centric component (e.g. going from 2-10yrs to 2-12yrs) of a paired vaccination programme that were evaluated as cost-effective. The colour of the arrow distinguishes willingness to pay values per vaccine dose that were less than £5 (red) or £5 and above (black). The green box contains the assessment of incremental expansion from the current low-risk coverage offered by the seasonal influenza vaccine policy for England.

most restrictive threshold vaccine dose price (£4.68) had minimal coverage in children (two year olds only), but included those aged 66 years and above (Fig 5(a)).

Similar features appearing from the inspection of paired strategies were apparent if adopting only one of the two coverage schemes. Considering a standalone young-age centric low-risk programme, we observe a peak in threshold vaccine dose price for the 2–6yrs strategy (£37.57). We witness an apparent dip in threshold price for programmes that include coverage of secondary school aged children, with strategies encompassing coverage of 2–12yrs and 2–14yrs returning threshold prices in the £15-£20 range (Fig 5(b)).

Moreover, for a sole elder-age centric low-risk vaccination programme, campaigns focusing resources on just the eldest ages had a sizeable return on each vaccine deployed (Fig 5(c)). Explicitly, the threshold vaccine dose price rose from £10.98 for low-risk coverage of those aged 90 and above, to £50.94 for low-risk coverage of those aged 98 and above. If wider age coverage is desired, covering the elder ages beginning above 50 years of age, we observed a regime were the threshold vaccine dose price lay below £10. In contrast, strategies including young adults, adolescents and children were found to have threshold vaccine dose prices between £10–£15 (though the total amount of additional vaccines required is greater, meaning a larger increase in absolute spend). Furthermore, the coverage offered by the current seasonal influenza vaccine policy for England to those at low-risk (young-age centric component for ages 2-10, elder-age centric component for those aged 65 and above) lay within the £10–£15 threshold vaccine dose price bracket. When we instead applied a 1.5% discounting rate for both costs and health effects (rather than a 3.5% discounting rate),we acquired quantitatively similar threshold dose prices for cost-effectiveness across all considered age coverage combinations (see Figs. H-I in S1 Text and S2 File).

The strong performance of a standalone young-age centric programme was also demonstrated for evaluations based on potential savings for a specified amount of additional vaccines administered over the entire time horizon. Under such a measure, standalone young-age centric programmes consistently outperformed standalone elder-age centric strategies (Fig 6, Fig. J in S1 Text).

**Optimal coverage of a low-risk vaccination programme for a fixed vaccine cost.** The apparent stark discrepancy in benefits derived from the young-age centric versus elder-age centric component of the low-risk vaccination scheme was also shown when we altered perspective to determine the optimal level of age coverage for a fixed cost of seasonal influenza vaccine.

Under our two differing assumptions of monetary worth of a QALY, we found the upper age bound of the young-age centric component to not exceed 65 years of age under any circumstance, even if there were no additional cost for procuring the vaccine. In addition, for the range of vaccine costs where a paired strategy was the top performer, the optimal age coverage for the young-age centric component of the programme usually included 0–20yrs. On the other hand, age coverage stemming from the elder-age centric aspect of the programme was very limited, with a typical coverage including those aged 98 years and above (Fig 7(a) and 7(b)).

We saw consistent behaviour when reperforming the analysis with the addition of a condition mandating that the elder-age component had to include, at the very least, all those aged 65 and above. Across each tested fixed vaccine dose price, the discerned optimal lower bound of age coverage for the elder-age component of the programme did not adjust from 64 years of age. Imposing the mandated age coverage condition also led to a narrowing in the range of influenza vaccine dose prices for which addition of a low-risk vaccination programme would be viable (Fig 7(c) and 7(d)).

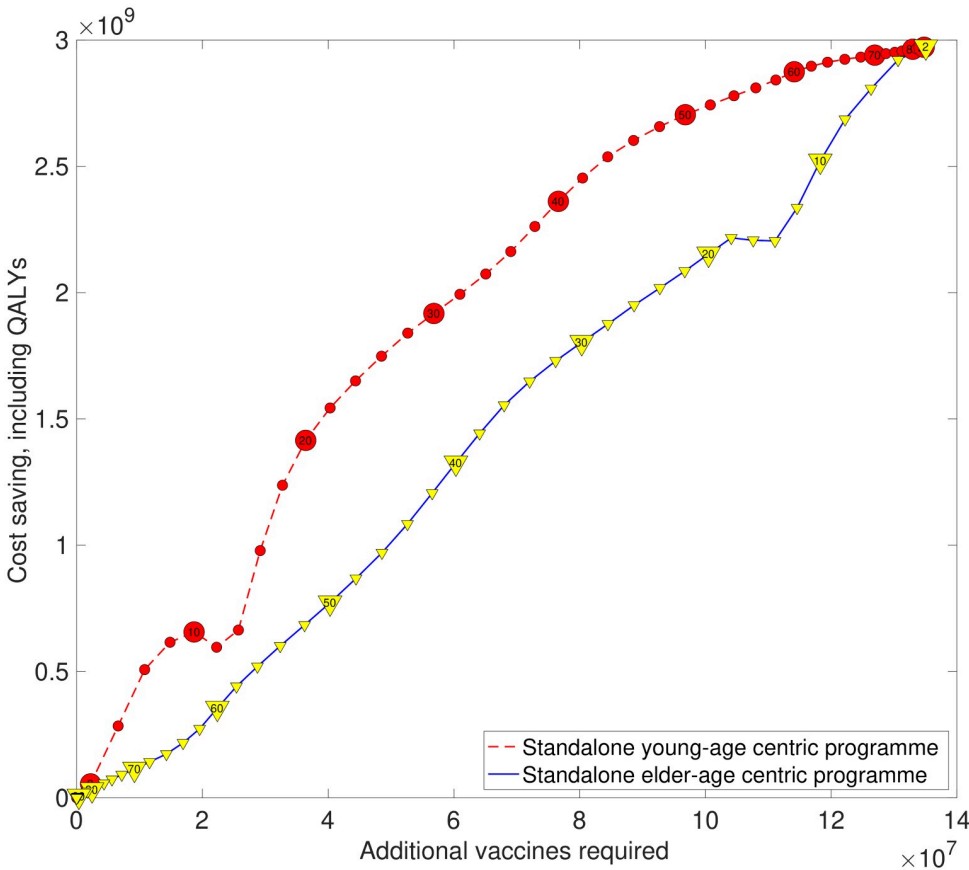

**Fig 6. Estimated savings against additional administered vaccines for alternative vaccine programmes (relative to an at-risk only vaccination programme).** The plotted points represent estimates for a single vaccine programme, with one QALY valued at £30,000 and the vaccine cost set at the threshold value that 90% of simulation replicates remain cost-effective. The red dashed line with circle markers corresponds to young-age centric strategies. Numerical values contained within the circle markers state the upper bound of age coverage represented by that particular young-age centric strategy (e.g. the marker containing "20" represents a strategy covering those between 2 to 20 years of age). The blue solid line with inverted yellow triangle markers corresponds to elder-age centric strategies. Numerical values contained within the inverted triangle markers express the lower bound of age coverage represented by that particular elder-age centric strategy (e.g. the marker containing "20" represents a strategy covering those aged 20 years old and above). Standalone young-age centric strategies perform well, forming a rough upper bound on potential savings. In contrast, standalone elder-age centric strategies typically display less effective performance.

**Incremental age expansion of a low-risk vaccination policy.**    Were a pre-existing vaccination policy to already contain one or both of a young-age centric or elder-age centric low-risk vaccination component (in addition to vaccination of those at-risk), we explored whether an incremental expansion in age coverage (corresponding to the inclusion of an additional two single year age brackets) would be deemed cost-effective. While we once more focus on the results produced under the uncertainty analysis setup (90% of simulations being cost-effective when a QALY is valued at £30,000) and a discount rate of 3.5%, we acquired similar outcomes using a cost-effectiveness threshold of £20,000 per QALY and a 1.5% discount rate. We provide a complete listing of threshold vaccine dose prices for all scenarios in S3 File.

To begin, we view from the perspective of expanding the young-age centric low-risk vaccination aspect of the programme. There are two noticeable boundaries where inclusion of additional ages was not evaluated to be cost-effective. The first boundary corresponded to switching from 2–10yrs coverage to 2–12yrs coverage. The second boundary was when

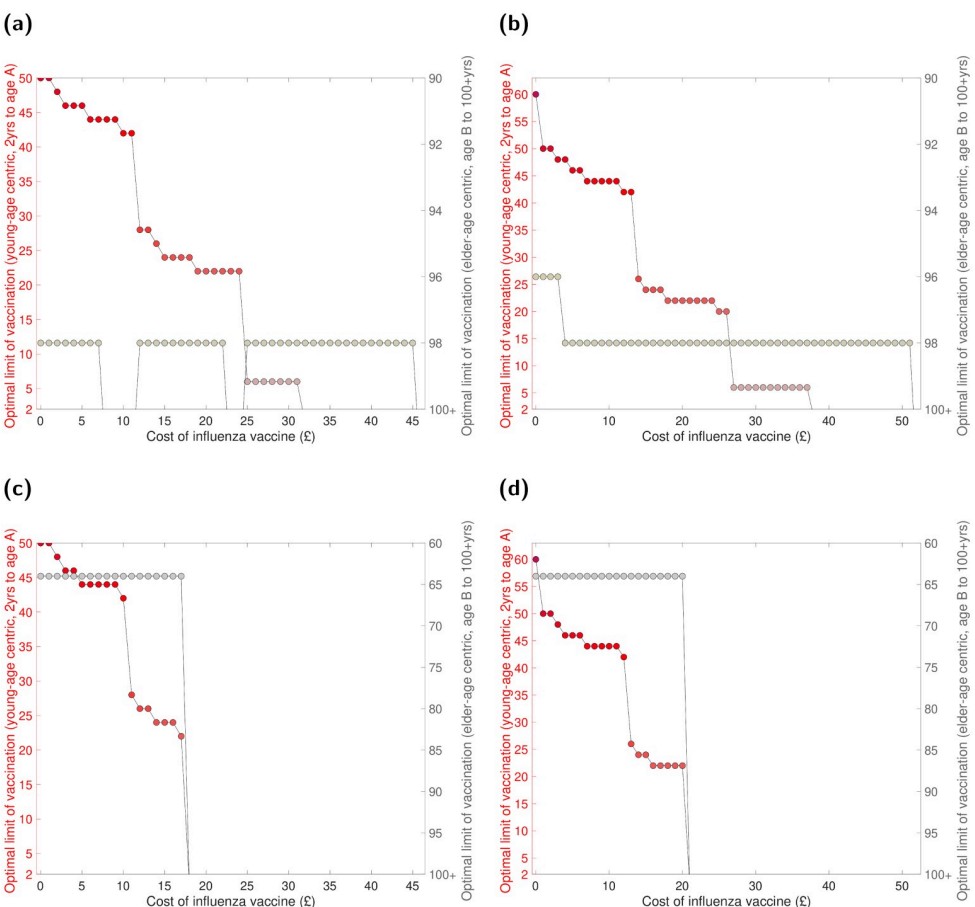

**Fig 7. Age coverage for each arm of the low-risk vaccination programme that returned the greatest monetary gains, given a specified cost of influenza vaccine.** Optimal age ranges were calculated at £1 increments for influenza vaccination, in each case under an assumption of 10% vaccine wastage and a £10 administration charge. Upper age bound quantities for the young-age centric programme and elder-age centric programme (per cost of influenza vaccine values) are indicated by the filled red circles and grey circles, respectively. Columns correspond to optimal age coverage assuming: **(a,c)** a £20,000 cost-effectiveness threshold per QALY, successfully attained by the most likely set of model parameters; **(b,d)** £30,000 cost-effectiveness threshold per QALY, with 90% of all parameter sets generating cost-effective results. The outcomes displayed in panels **(a,b)** had no mandated age coverage conditions; we found the upper age bound of the young-age centric programme to not exceed 65 years of age under any circumstance, whilst age coverage stemming from the elder-age centric aspect of the programme was usually limited or not required. In panels **(c,d)**, the elder-age arm of the low-risk vaccination programme had to include, at the very least, all those aged 65 and above. Amongst the programmes discerned as optimal (at each fixed vaccine dose price), none extended the lower age bound for the elder-age centric component below the age of 64. The mandated age coverage condition also narrowed the range of influenza vaccine dose prices for which the addition of a low-risk vaccination programme would be viable.

widening the programme to reach 50 year olds. Turning attention to expansion of the elder-centric low-risk vaccination programme, if the present vaccination policy included an elder-centric low-risk vaccination programme whose lower age bound was already below 65 years, then we see a general behaviour of incremental expansion being cost-effective if the cost of each additional vaccine was below a given threshold value (Fig 5(d)).

Combined, these characteristics illuminated regions of age space where, if the current vaccine programme were to be situated, you would not consider broadening age coverage. One such instance was if coverage in children was for 2–10 year olds, together with the lower edge of the elder-age centric component residing in either the 68–76 or 80–86 years of age interval

(Fig 5(d)). More generally, when coverage included children, adolescents and younger adults, the extra addition of elder adults was deemed not cost-effective.

## Discussion

Globally, seasonal influenza is an infection of grave public-health importance, given the economic cost and burden to human health afflicted by the condition [55]. In temperate regions such as England, during winter periods seasonal influenza inflicts a particularly stark stress on the health system [1]. With vaccination endorsed as an effective procedure to prevent seasonal influenza infection [2], policy makers have an ongoing interest in optimising cost-effectiveness of conjectured vaccination programmes. We contribute here an inspection of threshold vaccine dose prices (for England) to achieve cost-effectiveness under several differing paired age band low-risk vaccination schemes. Our outcomes suggest seasonal influenza vaccination programmes that target primary school aged children and young adults may present an opportunity to prevent a higher burden of disease at a lesser cost compared to vaccination of the elderly.

In order to procure reliable outputs from model simulations of conjectured vaccine policies, one first requires a disease transmission model calibrated to the available historical data. In this study, we used England specific surveillance data post the 2009 influenza pandemic to fit a parsimonious mechanistic model (including multiple strains, age-structure and immunity propagation) to seasonal-level data on strain competition. Our estimates for the immunity propagation associated parameters are in agreement with our findings from the simplified, non-age model structure [16]. Namely, the mechanism delivering an impact on the transmission dynamics was the modification to susceptibility due to natural infection in the previous influenza season, whilst there was no carry over of type B influenza cross-reactive immunity and little support for seasonal influenza vaccination stimulating long-term immunity responses. Once again, these conclusions corroborate previous immunological studies showing that infection with influenza virus can induce broader and longer-lasting protection than vaccination [34, 56, 57], and authenticate prior work signalling that vaccine-mediated immunity rapidly wanes [58].

Nevertheless, relative to the non-age structured model framework, the incorporation of age-structure did bring about minor quantitative discrepancies. Chiefly, under the modelling assumptions adopted within this study, the closest correspondence of model output to the empirical data was brought about when the propagation of immunity derived from natural infection was strengthened. In detail, when using an epidemiological model with age-structure, if infected by a given strain last influenza season we estimated an approximate 28% reduction in susceptibility to the current influenza season variant of that strain type, whilst the equivalent estimate when modelling the transmission dynamics with a non-age structured model was roughly 21%.

Our model fitting procedure resolved baseline seasonal influenza susceptibility to have a V-shaped structure, with susceptibility lower in adults than children, and greatest in the eldest. This structure married up with susceptibility patterns acquired from prior, England-specific, modelling work [10].

Relative differences in ascertainment probability inferred between influenza seasons resembled those from our previous analysis utilising a non-age structured model framework [16]. The greatest ascertainment probabilities were obtained for the 2017/18 influenza season. We conjecture that the propensity of those infected and symptomatic during the 2017/18 influenza season to consult a GP may have been abnormally raised as a result of moderate to high levels

of influenza activity being observed in the UK [18], whilst there was also a mismatch of the A (H3N2) component of the available vaccine towards the A(H3N2) strain in circulation [31].

Previous modelling studies applied to data from England revealed a V-shaped curve where ascertainment probability was higher in children (0–15yrs) and elderly adults (65+yrs) and twice as low in adults under 65 years old (16–64yrs) [10]. Our discerned ascertainment profile is in harmony with these outputs. Furthermore, given we modelled our ascertainment profile at a finer granularity, we uncovered a slight decrease in case ascertainment as age increased from 65 to 85 years of age. The relationship between the age an individual is afflicted with influenza illness and their propensity to consult a GP, particularly for elder ages, merits further scrutiny.

Our analysis of combinations of low-risk uptake amongst children, adolescents and the elderly revealed that schemes including children and adolescents had a higher allowable spend per vaccine dose (whilst maintaining overall programme cost-effectiveness versus the reference strategy of at-risk vaccination). These findings are reinforced by our sweep of the age space for proposed vaccination programmes, which unveiled maximisation of the threshold price per vaccine dose when coverage either spanned primary school aged children or reach also comprised young adults (with little coverage amongst the elder ages in either case).

We postulate these outcomes are due to the underlying contact structure. With school-age children considered to drive seasonal influenza transmission [59–61], a targeted paediatric programmes can contribute substantially towards founding herd immunity amongst the entire population. Meanwhile, a portion of young adults will be entering parenthood and therefore be at heightened risk of contracting illnesses from their young children. Additionally, young adults have more frequent contact with a broader portion of the general adult population (compared to young children), and therefore expanding vaccination into this age range provides its own contribution to establishing herd immunity.

The regions of age space with the largest threshold vaccine price encompassed elder-age centric programmes with a start age of 85 years and above. Notably, elder-age centric programmes beginning within the 65-75yrs age group had the least permitted expense per vaccine. A likely explanation for this occurrence is through our health economic assumptions; the order of magnitude for the likelihood of fatality (due to influenza infection) noticeably increases for those aged 85 years and above relative to younger ages. Thus, for those elder-age centric strategies beginning below 85 years of age, the QALYs gained through averted deaths per vaccine become less stark. The end result is less benefit per additional vaccine administered in order to cover additional ages. Another possible contributory factor is there being a bulge in the population structure for the 65-75yrs age group (resulting from the baby-boom generation).

The seeming preference, in terms of cost-effectiveness, for prioritising vaccine uptake amongst children (rather than the elderly) is in agreement with judgements from prior studies [9, 13]. In brief, Hodgson et al. [13] assessed the effect of different paediatric vaccine coverages on the effectiveness of the low-risk elderly and at-risk vaccine programmes in England and Wales. The authors established that a paediatric programme increasing its scope from pre-school-age children only to also include school-age children (with uptake consistent with other school-based programmes) would possibly raise significant uncertainty concerning the cost-effectiveness of a low-risk elderly vaccination programme. Similar conclusions have been obtained from cost-effectiveness analysis of seasonal influenza vaccination programmes in other countries. One such finding was borne out of an analysis for Japan; in comparison to a strategy using the empirically recorded vaccine uptake, a study by Tsuzuki et al. [9] predicted a vaccination programme targeting 90% uptake amongst children under 15 to have a much a larger epidemiological impact than those targeting higher vaccine uptake in the elderly.

This study may offer promising findings for decision making of seasonal influenza vaccination policy in England. However, we heed caution when interpreting the results due to the presence of several limitations.

It is important to stress that the utilised data were not completely sufficient to precisely capture the historical seasonal influenza epidemiology. In particular, virological strain composition data were not stratified by age group. Due to these constraints, for model fitting purposes we applied the population aggregated strain composition data to each single age group. However, there is an imminent prospect of this limitation being mitigated by planned work to link RCGP and PHE virology data (Simon de Luignan, personal communication). Incorporation of these age-stratified data would permit refinements to the parameter inference scheme to capture any heteorgeneities across ages.

A second age-dependent attribute warranting continued refinement is susceptibility to infection. For purposes of model parsimony, we enforced rigidity in the permitted form of the age-dependent susceptibility profile. By using four age classes, with younger adults (18-64 years of age) mandated to have lesser susceptibility compared to all other, we may have averaged over important heterogeneities. The impact of relaxing these assumptions merits further study. In tandem, the gathering of empirical data to inform age-dependent susceptibility profiles would be immensely beneficial with regards to reducing parameter uncertainty.

Another constraint related to the transmission model in our study is we assumed ascertainment probabilities to be constant over the course of an influenza season. In reality, contributory quantities to ascertainment likelihood may vary over an influenza season, in particular the propensity to consult a GP if suffering symptomatic illness. It is conceivable, for example, that the occurrence of an unexpectedly large seasonal epidemic may induce increased media coverage, with consequential changes in perception of the disease among the population where the outbreak is occurring.

In our proposed vaccination programmes, we enforced a simplifying assumption of having a consistent set of vaccine uptakes across all simulated influenza seasons. Nonetheless, we selected low-risk coverage of 60% for ages below 65 as we consider such coverage to be an attainable level in the present setting (based on recorded uptakes in recent influenza seasons amongst the elderly and the school-delivered paediatric programme [18, 19]).

With regard to our health economic parameters, we imposed a precise collection of age- and strain-specific conditions under which hospitalisation and mortality events could occur (i.e. for type B influenza cases, there were no mortality events (irrespective of age) and hospitalisations were permissible only for those aged under 10). Nevertheless, we contend our selected assumptions reflect a relevant evidence base.

Finally, we reiterate that the findings borne out from our cost-effectiveness analysis are estimates under a specific modelling framework and collection of assumptions. For example, compared to the circumstances presented throughout this manuscript, the presence of an effect such as negative vaccine interference in a subset of age groups could conceivably diminish the cost-effectiveness of alternate vaccine programmes. We advocate ongoing research into seasonal influenza immunity interactions across multiple influenza seasons to enhance biological understanding of seasonal influenza transmission dynamics, which will concurrently heighten the robustness of any subsequent health economic analysis.

In summary, we use our mathematical and health economic model to appraise conjectured seasonal influenza vaccination programmes for England that offer vaccination to all those low-risk individuals younger than a given age (but no younger than two years old) and all low-risk individuals older than a given age, while maintaining vaccination of at-risk individuals of any age. With regard to the low-risk portions of the population to target, our analysis predicts the largest benefits (in terms of cost-effectiveness from a healthcare provider perspective) to be

generated when prioritising vaccine uptake amongst primary school aged children and young adults. As such, our findings may be used to convey optimal target age groups for a seasonal influenza vaccination programme in England.

## Supporting information

**S1 Text. Supporting information for 'Optimising age coverage of seasonal influenza vaccination in England: A mathematical and health economic evaluation'.** This supplement consists of: (1) Data descriptions; (2) Complementary details of the epidemiological modelling approach; (3) Complementary details of the health economic modelling approach; (4) Additional results.
(PDF)

**S1 File. Threshold vaccine dose prices for low-risk uptake amongst paediatric and elder-age centric age groups.** Paediatric strategies covered the following age ranges: None, 4-10yrs (primary school only), 4-16yrs (primary and secondary school), 2-4yrs (pre-school only), 2-10yrs (pre-school and primary school aged children), and 2-16yrs (preschool, primary and secondary school). Per paediatric strategy, we considered five elder-age centric programmes, each having a varying breadth of age coverage: 50 years and above, 60 years and above, 70 years and above, 80 years and above, 90 years and above. Threshold vaccine dose prices reflect the value at which the programme becomes cost-effectiveness at a WTP of £20,000 per QALY, with 3.5% discounting for monetary costs and health effects. In all considered strategies, we assumed 10% vaccine wastage and an administration fee per deployed vaccine of £10.
(XLSX)

**S2 File. Threshold vaccine dose prices for the addition of low-risk vaccination programmes.** The file comprises of four sheets: (i) Willingness to pay at £20,000 per QALY and 3.5% discounting (single prediction using the most likely set of parameters); (ii) willingness to pay at £30,000 per QALY and 3.5% discounting (values at which 90% of simulations were cost-effective); (iii) willingness to pay at £20,000 per QALY and 1.5% discounting (single prediction using the most likely set of parameters); (iv) willingness to pay at £30,000 per QALY and 1.5% discounting (values at which 90% of simulations were cost-effective). In all considered strategies, we assumed 10% vaccine wastage and an administration fee per deployed vaccine of £10.
(XLSX)

**S3 File. Threshold vaccine dose prices for incremental age expansion of low-risk vaccination programme.** The file comprises of four sheets: (i) Willingness to pay at £20,000 per QALY and 3.5% discounting (single prediction using the most likely set of parameters); (ii) willingness to pay at £30,000 per QALY and 3.5% discounting (values at which 90% of simulations were cost-effective); (iii) willingness to pay at £20,000 per QALY and 1.5% discounting (single prediction using the most likely set of parameters); (iv) willingness to pay at £30,000 per QALY and 1.5% discounting (values at which 90% of simulations were cost-effective). In all considered strategies, we assumed 10% vaccine wastage and an administration fee per deployed vaccine of £10.
(XLSX)

## Acknowledgments

We thank Rachel Byford, Ana Correa, Chris McGee, Julian Sherlock and Sameera Pathiranne-helage for their collective contribution towards the production of the RCGP RSC data extract utilised in this study. We express thanks to patients who don't opt out of data sharing, and

practices who are members of the RCGP RSC network. We acknowledge Computerised Medical Record systems suppliers: EMIS, TPP Systm One, In Practices Systems and Wellbeing for their collaboration facilitating RCGP RSC data extraction. Public Health England are the principle funders of RCGP RSC. We thank Tom Irving and colleagues at Department of Health and Social Care for helpful discussions. We acknowledge Warren Tennant for constructive feedback on the manuscript.

## Author Contributions

**Conceptualization:** Edward M. Hill, Stavros Petrou, Matt J. Keeling.

**Data curation:** Henry Forster, Simon de Lusignan, Ivelina Yonova.

**Formal analysis:** Edward M. Hill.

**Funding acquisition:** Stavros Petrou, Matt J. Keeling.

**Investigation:** Edward M. Hill, Matt J. Keeling.

**Methodology:** Edward M. Hill, Matt J. Keeling.

**Software:** Edward M. Hill.

**Supervision:** Edward M. Hill, Stavros Petrou, Matt J. Keeling.

**Validation:** Edward M. Hill, Matt J. Keeling.

**Visualization:** Edward M. Hill, Matt J. Keeling.

**Writing – original draft:** Edward M. Hill.

**Writing – review & editing:** Edward M. Hill, Stavros Petrou, Henry Forster, Simon de Lusignan, Ivelina Yonova, Matt J. Keeling.

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
