## [Decision Letter · Decision Letter 0]

7 May 2020

Dear Dr Hill,

Thank you very much for submitting your manuscript "Optimising age coverage of seasonal influenza vaccination in England: A mathematical and health economic evaluation" for consideration at PLOS Computational Biology.

As with all papers reviewed by the journal, your manuscript was reviewed by members of the editorial board and by several independent reviewers. In light of the reviews (below this email), we would like to invite the resubmission of a significantly-revised version that takes into account the reviewers' comments.

Please do make the code available for subsequent rounds of revision.

We cannot make any decision about publication until we have seen the revised manuscript and your response to the reviewers' comments. Your revised manuscript is also likely to be sent to reviewers for further evaluation.

Sincerely,

Benjamin Althouse

Associate Editor

PLOS Computational Biology

Virginia Pitzer

Deputy Editor

PLOS Computational Biology

Please do make the code available for subsequent rounds of revision.

Reviewer's Responses to Questions

**Comments to the Authors:**

Reviewer #1: The authors use a transmission dynamic model of influenza to evaluate health and economic effects of extending influenza vaccination beyond high risk individuals to different age groups of ‘low risk’ individuals in England. The question of optimal age coverage of influenza vaccination is not novel, but a specific area where this paper differentiates itself from previous work more sophisticated treatment of questions related to (partial) immunity conferred in previous seasons by natural infection and/or vaccination. Of note, this paper extends the model previously published in this journal in 2019 to include age structure which served to introduce the feature of carryover of immunity from season to season (ref 16 of current paper). The model is complex, attempting to simultaneously account for a) strain diversity, b) population age structure, c) infection and vaccination history, as well as d) observation processes mapping true dynamics onto surveillance data, but the writing is clear and (for the most part) assumptions are clearly stated.

Major comments:

1) This is an impressive piece of research. The authors have extended their previous model to address policy relevant questions about the health and economic consequences of expanding seasonal influenza vaccination by age groups. This is an important contribution to the literature.

2) A major challenge for the authors is to determine where to focus their attention given the number of reportable results included in the paper. They have focused their main conclusions on health and economic consequences of competing vaccination policies, but there are many important results which the model provides some additional insight about, including inference on strain specific reproductive numbers and strain-specific partial immunity afforded by previous infections. The following comments in this section reflect my concern that the complexity of the model and richness of the results, also means that readers will have numerous questions about the importance of underlying assumptions for the conclusions being drawn (even if these are not featured as key conclusions), not all of which can probably be reasonably subjected to sensitivity analyses. I would suggest the authors may better help a reader understand which of these types of assumptions are most critical for driving the conclusions that they draw.

3) The model is complex and requires numerous assumptions which often not verifiable with available data and may be important for inference. Can the authors help guide the reader better about how these might affect results and how they should be viewed by readers? A partial list includes:

a. Assumptions needed to estimate strain specific vaccine efficacy (Table 1, Table S1).

b. Line 89: Assumption that strain distribution is maintained across age groups

c. Line 170: Assumptions about how strain-specific immunity afforded by vaccination carries over between seasons.

d. Line 202: While the motivation for assumption about average susceptibility within age bands is reasonable – any sensitivity analysis on this? Age band from 18-64 is very wide. Does this result in averaging over important heterogeneities?

4) Lines 212-220: I am struggling to fully understand procedures for model initialization. How did authors establish age-group specific residual immunity f(h,m) at beginning of simulations?

5) Ascertainment probabilities – for those >100y which serve as reference group from which all other age group ascertainment probabilities are calculated is very low. The ascertainment probabilities seems terribly small, in the majority of cases well <1%. Is this plausible? Am I interpreting Table 6 correctly?

6) Figure comments

a. Fig 5 (and similar figures in the supplement) are too difficult to read. In particular the arrowheads are very hard to make out.

b. Fig 6 (and similar figures in supplement) – I am really struggling to interpret these! The size shapes and shapes of the figures are difficult for me to see at full size. I fear I am missing the conclusions here. Can simpler figures be made, even if more panels are required to show all the information?

7) Are the authors making their code available?

Minor comments:

1) Line 33: Claim that previous models guiding England policy were calibrated to pre-2009 data begs the question of whether this is materially important for the insights generated here. Of course, it is good that the authors have calibrated their model to more recent data, but this claim implicitly suggests that a model similar to theirs calibrated to older data might result in different conclusions. Is that actually true? If not, I would suggest that they remove this from the Introduction so that the main advance here remains focused on their treatment of mechanisms of carry-over immunity from season to season .

2) Line 225: Claim that health outcomes only rather than epidemiological processes are dependent upon risk group membership seems too string a statement. I am not quibbling with the authors’ approach here, as indeed risk group most profoundly affects health outcome condition on infection status, but the assertion that risk group is entirely divorced from transmission dynamics is probably too strong a statement to make.

3) Is there a good reason to retain the labels for the 5 different pediatric vaccination schemes (Prim, PrimSec, etc)? Why not just label these as the age groups targeted which is nearly as concise and is easier for readers as they don’t need to remember anything? Maybe I am missing an obvious rationale for this…

Reviewer #2: In this manuscript entitled “Optimising age coverage of seasonal influenza vaccination in England: A mathematical and health economic evaluation”, Hill et al. developed an epidemiological model for all 4 influenza strains and paired it with a between-seasons map that allowed them to propagate immunity from one season to the next. They combined this with a cost-effectiveness analysis for influenza vaccination among low-risk individuals >2 years old.

Major concerns

This manuscript has some very interesting ideas and the analysis presented is certainly very thorough. However, after reading the manuscript many times, I still don’t fully understand key points of the article:

1) Unless I completely misunderstood this manuscript, the authors are comparing many strategies to a “high-risk only strategy”, that would not include all people over 65 years old or all primary school aged children. However, according to this website:

https://www.nhs.uk/conditions/vaccinations/flu-influenza-vaccine/

People over 65 years old, children aged 2 and 3 and children in primary school are already getting the vaccine. Then it is unclear to me what the authors are testing against. Why would it make sense to use a “high-risk only” strategy as a baseline if this does not correspond to the current guidelines of the country?

If I misunderstood what you are comparing, then more explanations are needed.

If I understood this correctly, your baseline strategy is not the one currently used by the UK government. Unless there is a debate about contracting (so vaccinating less people) the status quo policy, then the article needs a very strong rationale about why this is a relevant comparison.

In particular, it either needs to state very clearly why contracting the vaccination for the elderly is a valuable exercise (considering how unpopular would be to take away vaccination from older people, who are the higher-risk group for influenza and the ethical implications of doing this) or focus on studying expanding the vaccination program to age groups not currently offered vaccination.

Given the fact that the UK already offers vaccination both to primary school aged children and the elderly, then I believe that the most relevant scientific question might be if it would be beneficial to expand the current recommendations to expand the age vaccinated (either by expand the children-centric part of reducing the age to start vaccinating the elderly groups). This is already somewhat covered in the manuscript (except that the comparison is against a high-risk strategy only), but it is buried among many other results. It might be better to concentrate on this and all the related results for this (expanding children only, older age groups only, both, optimal level of age coverage for a fixed cost of seasonal influenza vaccine, threshold vaccine dose price, etc)

Maybe a table/figure with which groups are covered under the baseline strategy and which ones are tested (in age bands, one obviously would not want to include the >1000 strategies tested) could be very helpful.

2) Overall, the sections regarding the implementation of the vaccination strategies are very confusing. I believe it will strongly benefit from a thorough rewrite, perhaps with more tables and figures summarizing the main points.

3) I am intrigued about the value of the Prior influenza season vaccine efficacy propagation being 0. More explanation is needed. Does that mean that being vaccinated the previous year did not provide any additional protection? Also, did the authors allowed this parameter to take negative values? There is a big debate now in the field about the topic of negative interference. I think this parameter should be allowed to be negative. Considering the that the authors did a cost-effectiveness analysis, it is conceivable that the vaccine would be less cost-effective if this parameter is negative, so I believe this needs to be explored/discussed.

4) The values of the R0 obtained for the A strains are really high. How was R0 computed? I could not find that information anywhere in the article. At any given season, there is a percentage of the population that should be immune to the circulating strain, as it is indeed indicated by the authors’ estimate on the “modification to susceptibility due to natural infection”. Hence, I am wondering if the authors should compute R_effective for each season in addition to R0, since this will give a more realistic idea of what each season is experiencing in terms of transmission.

5) Goodness of fit (paragraphs l. 456-473) It seems that the only comparison between the data and the 100 simulations obtained through optimization is a visual comparison given in figure 3. I think a more quantitative assessment of the model outcomes vs data is needed. For instance, what how does the median and inter-quartiles of the simulations compare to the data? It seems that for some years and some age groups, the model fits pretty well the data (eg. 2014-2015) but not so much for others (eg 85+ for 12/13 season or 15/16) and in general it vastly overestimates either B-strain when the other one was predominant. While there is a comment in the text about the older age group and B-subtype not being fit properly, I believe a more thorough/quantitative analysis of the fit is warranted.

6) Over which years were the vaccination strategies computed to assess their cost effectiveness? At some point of the text (l.285) it is stated that the calibration was done from 2012-2018, at some other point the authors talk about the baseline vaccination strategy and how they defined it in 2009/10 (l. 291) and finally when the authors define the time horizon of the model they come back to 2012-2018 (l.407).

7) I don’t quite understand how the cost-effectiveness analysis was carried out: was this a retrospective analysis of what would have been the most cost-effective program for those years? If so, I think this needs a strong rationale of why carrying out a retrospective analysis of what would have been cheaper/more effective would be important today. Were the interventions modeled in those years used as some sort of average to determine the cost of each of these alternative strategies and to decide which ones would be the more cost effective in the future? The paper as it stands is very hard to follow and I could not understand what was done.

Minor concerns:

Cost-effectiveness should be cost-effective in l.484

Lines 582-590, the text refers to figure 3 but given what figure 3 is, I believe this is not the correct figure (making the paragraph difficult to follow)

l.433-435 “error measure value below 30,000” what does that mean? Where does the 30,000 come from? More explanation needed

I wonder if there is a better way to represent the information contained in figures 5 and 6 that are very hard to understand.

Paragraphs starting on lines 305 and 326 are unclear and would benefit from some re-writing.

**Have all data underlying the figures and results presented in the manuscript been provided?**

Reviewer #1: Yes

Reviewer #2: Yes

PLOS authors have the option to publish the peer review history of their article (what does this mean?). If published, this will include your full peer review and any attached files.

Reviewer #1: No

Reviewer #2: No
---

## [Decision Letter · Decision Letter 1]

20 Aug 2020

Dear Dr Hill,

We are pleased to inform you that your manuscript 'Optimising age coverage of seasonal influenza vaccination in England: A mathematical and health economic evaluation' has been provisionally accepted for publication in PLOS Computational Biology.

Best regards,

Benjamin Althouse

Associate Editor

PLOS Computational Biology

Virginia Pitzer

Deputy Editor

PLOS Computational Biology

Reviewer's Responses to Questions

**Comments to the Authors:**

Reviewer #1: The authors have been responsive and made several changes which improve the clarity of the manuscript. I still question the ascertainment probabilities estimated, but appreciate the authors response to the query and acknowledge there are few data to help here.

Reviewer #2: This manuscript was greatly improved!

**Have all data underlying the figures and results presented in the manuscript been provided?**

Reviewer #1: Yes

Reviewer #2: Yes

PLOS authors have the option to publish the peer review history of their article (what does this mean?). If published, this will include your full peer review and any attached files.

Reviewer #1: No

Reviewer #2: No

---

## [Editor Report · Acceptance letter]

30 Sep 2020

PCOMPBIOL-D-20-00486R1 

Optimising age coverage of seasonal influenza vaccination in England: A mathematical and health economic evaluation

Dear Dr Hill,

I am pleased to inform you that your manuscript has been formally accepted for publication in PLOS Computational Biology. Your manuscript is now with our production department and you will be notified of the publication date in due course.

With kind regards,

Matt Lyles
